# A Latent Diffusion Model for Protein Structure Generation

## Abstract

Proteins are complex biomolecules that perform a variety of crucial functions within living organisms. Designing and generating novel proteins can pave the way for many future synthetic biology applications, including drug discovery. However, it remains a challenging computational task due to the large modeling space of protein structures. In this study, we propose a latent diffusion model that can reduce the complexity of protein modeling while flexibly capturing the distribution of natural protein structures in a condensed latent space. Specifically, we propose an equivariant protein autoencoder that embeds proteins into a latent space and then uses an equivariant diffusion model to learn the distribution of the latent protein representations. Experimental results demonstrate that our method can effectively generate novel protein backbone structures with high designability and efficiency.

## 1 Introduction

The discovery of novel proteins (Anand & Huang, 2018; Eguchi et al., 2022; Anand et al., 2019; Sabban & Markovsky, 2020; Luo et al., 2022; Shi et al., 2022) is crucial in bio-medicine (Liu et al., 2021b; 2022a;b; Wang et al., 2022a;b; Liu et al., 2021a) and materials (McMillan et al., 2019; Yan et al., 2022). Recently, instead of generating novel protein sequences (Wu et al., 2021; Anishchenko et al., 2021; Ferruz et al., 2022; Repecka et al., 2021; Hawkins-Hooker et al., 2021; Madani et al., 2020; Nijkamp et al., 2022; Karimi et al., 2020) and then predicting their corresponding structures, Trippe et al. (2022) and Wu et al. (2022a) propose to directly generate protein structures using diffusion models, due to the impressive modeling power and generation quality of diffusion models (Ho et al., 2020; Song et al., 2020; Xu et al., 2021; Jing et al., 2022; Rombach et al., 2022) for images and small molecules. However, generating 3D protein structures is a more challenging task because of their complex geometric structures and vast exploration space. Additionally, as the modeling space increases, the cost of time and computational resources required to train and sample from diffusion models also increases significantly.

There are attempts to reduce the modeling space in the image and small molecule domain for diffusion models. Stable Diffusion (Rombach et al., 2022) combines a pretrained image autoencoder and a latent diffusion model to reduce the modeling space for large images. However, there are currently no robust and powerful 3D graph autoencoders and latent diffusion models for 3D protein structures. Torsional Diffusion (Jing et al., 2022) only focuses on torsional angles and employs RDKit (Landrum et al.) predictions for bond lengths and bond angles, as the distributions of bond angles and lengths are highly confined in small molecules. But this assumption does not hold for protein structures.

In this paper, we reduce the diffusion modeling space of complex 3D protein structures by integrating a 3D graph autoencoder and a latent 3D diffusion model. To achieve this, the following challenges are addressed: (1) ensuring rotation and reflection equivariance in the autoencoder design, (2) accurately reconstructing intricate connection information in 3D graphs during decoding, and (3) developing a specialized latent diffusion process for 3D protein latent representations, including position and node latent representations. In the following sections, we first recap the background and related works for protein backbone structure generation and diffusion models in Sec. 2, and then show in detail how we address the above challenges in Sec. 3. The efficiency and ability to generate novel protein backbone structures of our proposed method are demonstrated in Sec. 4.

## 2 Background and Related Work

### 2.1 Protein Backbone Structure Generation

Protein backbone generation aims to generate novel protein backbone structures by learning from real data distributions. To this end, a mapping between known distributions, such as a Gaussian, and the real data distribution, which is high dimensional and sparse, needs to be constructed. Since protein global geometric structures are mainly determined by backbones, the generation of protein structures can be simplified to the generation of backbones consisting of a sequence of amino acids and their corresponding positions. Following ProtDiff (Trippe et al., 2022), we use the positions of alpha carbons to represent amino acid positions. The protein backbone structure is then represented by

$$\mathcal{S} = \{(\boldsymbol{x}_i, a_i)\}_{i=1}^n, \tag{1}$$

where $\boldsymbol{x}_i \in \mathbb{R}^3$ denotes the 3D position of alpha carbon in the $i$-th amino acid, and $a_i \in \{k | 1 \leq k \leq 20, k \in \mathbb{Z}\}$ denotes the corresponding amino acid type.

Instead of modeling amino acid types and alpha carbon positions together, previous studies (Trippe et al., 2022) have shown that it is better to decompose the whole generation process into two stages as $p(\boldsymbol{x}, \boldsymbol{a}) = p(\boldsymbol{a}|\boldsymbol{x})p(\boldsymbol{x})$, where $\boldsymbol{x} = [\boldsymbol{x}_1, \boldsymbol{x}_2, \cdots, \boldsymbol{x}_n]$, and $\boldsymbol{a} = [a_1, a_2, \cdots, a_n]^T$. Specifically, the positions of alpha carbons are first generated, and the corresponding amino acid types are predicted using pretrained inverse folding models such as ProteinMPNN (Dauparas et al., 2022).

### 2.2 Denoising Diffusion Probabilistic Models

As a powerful class of generative models (Luo et al., 2021; Liu et al., 2021c; Luo & Ji, 2022), denoising diffusion probabilistic models (DDPM) (Ho et al., 2020) solve the Bayesian inverse problem of deriving the underlying data distribution (posterior) $p_{\text{data}}(\boldsymbol{z})$ by establishing a bijective mapping between given prior distributions and $p_{\text{data}}(\boldsymbol{z})$. We review the background of DDPM here following the adopted conventions of ScoreSDE (Song et al., 2020). To enable faithful generation based on $p_{\text{data}}(\boldsymbol{z})$ by sampling simpler prior distributions, a discrete Markov chain is employed to gradually diffuse inputs as a map from given training data into random noise, for example, following multivariate normal (Gaussian) distributions. For every training sample $\boldsymbol{z}_0 \sim p_{\text{data}}(\boldsymbol{z})$, DDPMs consider a sequence of variance values $0 < \beta_1, \beta_2, \ldots, \beta_N < 1$ and construct a discrete Markov chain $\{\boldsymbol{z}_0, \boldsymbol{z}_1, \ldots, \boldsymbol{z}_N\}$, where $p(\boldsymbol{z}_i|\boldsymbol{z}_{i-1}) = \mathcal{N}(\boldsymbol{z}_i; \sqrt{1-\beta_i}\boldsymbol{z}_{i-1}, \beta_i\mathbf{I})$. Based on this, we obtain $p(\boldsymbol{z}_i|\boldsymbol{z}_0) = \mathcal{N}(\boldsymbol{z}_i; \sqrt{\alpha_i}\boldsymbol{z}_0, (1-\alpha_i)\mathbf{I})$, where $\alpha_i = \prod_{t=0}^i (1-\beta_t)$. Hence, a sequence of noise scales can be predefined such that $\alpha_N \to 0$ and $\boldsymbol{z}_N$ is approximately distributed according to $\mathcal{N}(\mathbf{0}, \mathbf{I})$. For the reverse mapping from $\mathcal{N}(\mathbf{0}, \mathbf{I})$ to $p_{\text{data}}(\boldsymbol{z})$, a reverse Markov chain is parameterized as $p_\theta(\boldsymbol{z}_{i-1}|\boldsymbol{z}_i) = \mathcal{N}(\boldsymbol{z}_i; \mu_\theta(\boldsymbol{z}_i, i), \beta_i\mathbf{I})$, where $\mu_\theta(\boldsymbol{z}_i, i) = \frac{1}{\sqrt{1-\beta_i}}(\boldsymbol{z}_{i-1} - \frac{\beta_i}{\sqrt{1-\alpha_i}}\mathbf{s}_\theta(\boldsymbol{z}_i, i))$. The reverse diffusion model $\mathbf{s}_\theta$ is trained with a re-weighted evidence lower bound (ELBO) as below

$$\boldsymbol{\theta}^\star = \text{argmin}_{\boldsymbol{\theta}} \mathbb{E}_{t, \boldsymbol{z}_0, \boldsymbol{\sigma}}[\|\boldsymbol{\sigma} - \boldsymbol{s}_{\boldsymbol{\theta}}(\sqrt{\alpha_t}\boldsymbol{z}_0 + \sqrt{1-\alpha_t}\boldsymbol{\sigma}, t)\|^2], \tag{2}$$

where $\boldsymbol{\sigma} \sim \mathcal{N}(\mathbf{0}, \mathbf{I})$. After $\mathbf{s}_\theta$ is trained, the reverse sampling process is conducted by first sampling from $\boldsymbol{z}_T \sim \mathcal{N}(\mathbf{0}, \mathbf{I})$ and then updating from time $N$ to time 0 by the estimated reverse Markov chain

$$\boldsymbol{z}_{t-1} = \frac{1}{\sqrt{1-\beta_t}}(\boldsymbol{z}_t - \frac{\beta_t}{\sqrt{1-\alpha_t}}\boldsymbol{s}_{\boldsymbol{\theta}}(\boldsymbol{z}_t, t)) + \sqrt{\beta_t}\boldsymbol{\sigma}. \tag{3}$$

### 2.3 Related Work

**Diffusion Models for Protein Structure Generation**. Recent research (Anand & Achim, 2022; Wu et al., 2022a; Trippe et al., 2022; Lee & Kim, 2022; Watson et al., 2022; Ingraham et al., 2022) has been exploring the use of diffusion models to generate novel protein structures, building on the successes of diffusion models in other areas such as images (Ho et al., 2020; Song et al., 2020) and small molecules (Xu et al., 2021; Jing et al., 2022; Hoogeboom et al., 2022). Among them, ProtDiff (Trippe et al., 2022) focuses on generating protein backbone structures by determining the positions of alpha carbons, while FoldingDiff (Wu et al.,

2022a) represents protein backbone structures using bond and torsion angles and applies a sequence diffusion model to generate new backbone structures. Anand & Achim (2022) attempts to generate the entire protein structure by using three separate diffusion models to generate alpha carbon positions, amino acid types, and side chain rotation angles sequentially, but the joint modeling performance is relatively low. Additionally, Lee & Kim (2022) proposes to diffuse 2D pairwise distances and angle matrices for amino acid residues, but further optimization using Rosseta minimization (Yang et al., 2020) is needed.

It is notable that, while developing our method, two recent works RFdiffusion (Watson et al., 2022) and Chroma (Ingraham et al., 2022) have been developed that enable generating long proteins with very high quality. RFdiffusion takes advantage of the powerful protein structure prediction model, RoseTTAFold (Baek et al., 2021), to achieve remarkable results on many generation tasks. RFdiffusion pretrains RoseTTAFold on the protein structure prediction task and then finetunes on generative tasks. And RFdiffusion only demonstrates the effectiveness of generating proteins only when using pretrained weights. Chroma uses a correlated diffusion process to transform protein structures into random collapsed polymers and encode the chain and radius of gyration constraints by a designed covariance model. In this way, Chroma can model the target distribution more efficiently by preserving some basic structures in proteins.

Despite the success of protein backbone structure generation (Anand & Achim, 2022; Wu et al., 2022a; Trippe et al., 2022; Lee & Kim, 2022; Watson et al., 2022; Ingraham et al., 2022), the modeling space of diffusion models is vast and increases exponentially with the number of amino acids considered.

**Decreasing Modeling Space for Protein Structure**. The modeling space for protein structure generation is reduced in several ways. ProtDiff (Trippe et al., 2022) only considers the positions of alpha carbons, while FoldingDiff (Wu et al., 2022a) represents protein backbone structures using bond and torsion angles and omits bond lengths to decrease the modeling space. Torsional Diffusion (Jing et al., 2022) uses RDKit-generated bond lengths and angles and only diffuses the torsional angles for the conformer generation of small molecules, but it is not applicable for protein structures.

Recently, the impressive generative capability of Stable Diffusion (Rombach et al., 2022) in the image domain has attracted significant attention. By integrating a pre-trained image autoencoder with latent diffusion models, Stable Diffusion reduces the modeling space of large images and improves the generative power of image diffusion models. However, 3D geometric graphs for protein structures are different from images, and even though there are some equivariant networks for modeling interatomic potentials (Batzner et al., 2022) or predicting protein binding sites (Zhang et al., 2023), no robust 3D equivariant protein autoencoders and 3D latent diffusion models for protein structures have been proposed yet.

## 3   Method

In this section, we introduce our LatentDiff for generating protein backbone structures. We first illustrate the motivation for reducing modeling space in Section 3.1. Then, We describe the design of our equivariant protein autoencoder in Section 3.2, and next the latent space diffusion model in Section 3.3. We present the overall generation process in Section 3.4.

### 3.1   Motivation of Reducing Modeling Space

In this section, we describe the motivation for designing a protein autoencoder to reduce modeling space in terms of modeling difficulty and parallel sampling efficiency, respectively. An important motivation for reducing modeling space through downsampling is that it can make the diffusion model easier to learn the desired distribution, as the modeling capacity of diffusion models has a direct relationship with the size of their modeling space. By decreasing the modeling space, we aim to focus the generative model's attention on a more condensed space that is relevant to the original protein structure space. This reduction in complexity allows the model to more effectively learn and capture the underlying distribution of protein structures, resulting in improved generation quality.

Moreover, a smaller modeling space helps address the challenge of high dimensionality and sparsity that is prevalent in protein structure data compared with small molecules. The vast space of possible protein

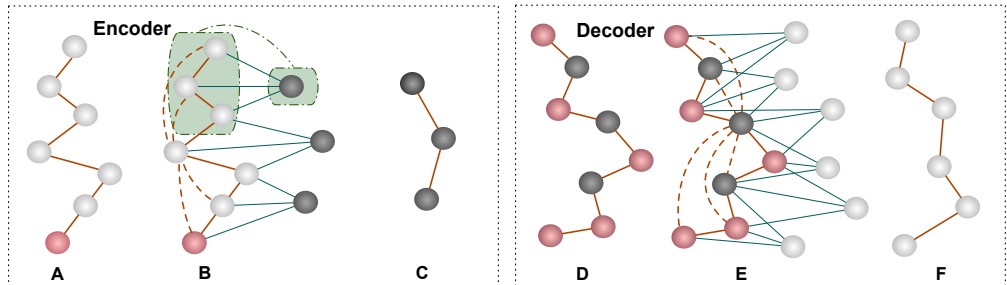

Figure 1: Autoencoder network structure for proteins. Step A, B, and C denote the Encoder network. A. Augmented input protein sequence (white) with padding (red node), similar to image padding. B. (1) Edge building: create a fully connected graph (limited edges shown for simplicity) on the padded sequence; (2) Graph Expansion: introduce new nodes (black) with specific connections according to the 1D-CNN convention. C. Compressed sequence (in latent space). Steps D, E, and F denote the Decoder network. D. Padding latent sequence for upsampling (similar to padding operation in image transpose convolution). E. Edge building and Graph Expansion are similar to B. F. Reconstructed protein chain.

conformations presents a considerable challenge for generative models to learn from limited training data. By narrowing down the modeling space, we provide the generative model with a more manageable and structured latent search space, enabling it to learn the essential features and patterns of protein structures more efficiently.

Another advantage of using protein autoencoder to reduce modeling space is that generation in latent space can improve memory efficiency as the latent space is much smaller than the protein space. So for the same amount of GPU memory, more proteins can be sampled in latent space than in protein space. In practice, it requires sampling a large amount of proteins in the screening procedure, so high throughput sampling is desired. In this sense, parallel sampling in latent space could demonstrate significant efficiency improvement. More experiments on parallel sampling efficiency can be found in Section 4.7.

### 3.2 Equivariant Protein Autoencoder

We first introduce our equivariant autoencoder that helps reduce the protein design space. To design such an autoencoder, we identify some constraints and the uniqueness of protein backbones. First, $C_\alpha$ atoms in protein backbones have a fixed order due to the sequential nature of amino acid sequences. In general, downsampling or upsampling of sequence data can be achieved by 1D convolutional neural networks (CNNs). Also, since $C_\alpha$ atoms form a chain structure that could be preserved during upsampling, we don't need to reconstruct edge connections like traditional graph autoencoder. Second, despite the sequence representation of protein backbones, they also possess 3D geometries, which require equivariance during the downsampling and upsampling stages. Traditional CNN cannot meet this equivariant requirement, but graph neural networks (GNNs) are capable of dealing with this challenge. Based on these observations, we propose a novel equivariant protein autoencoder that considers both the amino acid sequence and 3D graph information of protein backbones.

**Overview.** In the equivariant protein autoencoder, we first downsample protein to smaller sizes and upsample the latent graph to reconstruct the original protein. There are four steps within each downsampling and upsampling layer, namely **sequence padding**, **edge building**, **graph expansion**, and **equivariant message passing**. The first three steps are used to construct a graph that contains the input nodes and initialized downsampling or upsampling nodes in the current layer. After the message passing, only updated downsampling or upsampling nodes will be kept as input in the next layer for further downsampling or upsampling operation. In the following, we describe the network input and details of one downsampling layer. The upsampling layer shares the exact same steps except for sequence padding, which we will also introduce in the sequence padding section.

**Network Input.** For a protein backbone structure $\mathcal{S}$, we move the structure to the zero centroid in order to make the model avoid capturing translational equivariance. Then we will augment the protein to a fixed length $m$ to simplify the remaining operations in the network. So $m$ is the maximum protein length that we can generate, and we choose $m$ as 128 in this work. The augmented protein is shown as the white part in Figure 1.A. Specifically, we append $m - n$ extra nodes to the end of the protein sequence. Each extra node is assigned a zero position and the same node type. And we denote the augmented protein structure as $\mathcal{S}_{\text{aug}} = (\boldsymbol{X}, \boldsymbol{H})$, where $\boldsymbol{X} \in \mathbb{R}^{3 \times m}$ and $\boldsymbol{H} \in \mathbb{R}^{d \times m}$ are node positions and node feature vectors respectively. For $\boldsymbol{X}$, the first $n$ columns $\{\boldsymbol{x}_i\}_{i=1}^{n}$ denote the positions of all $C_\alpha$ atoms in the original protein and the last $m - n$ columns $\{\boldsymbol{x}_i\}_{i=n+1}^{m}$ denote the zero positions of extra nodes. Each node feature vector $\boldsymbol{h}_i \in \mathbb{R}^d$ in $\boldsymbol{H}$ is a $d$-dimensional type embedding indicating the corresponding node type. Then the preprocessed $\mathcal{S}_{\text{aug}}$ is the input to the first downsampling layer.

**Sequence Padding.** Similar to padding in image convolution, within each layer, we first need to pad the augmented protein sequence $\mathcal{S}_{\text{aug}}$ before downsampling or upsampling the sequence in order to obtain an output with the desired size. Let's assume that we have $k$ nodes after sequence padding. Denote the padded sequence as $\mathcal{S}_{\text{pad}} = (\boldsymbol{X}_{\text{pad}}, \boldsymbol{H}_{\text{pad}})$, where $\boldsymbol{X}_{\text{pad}} \in \mathbb{R}^{3 \times k}$ and $\boldsymbol{H}_{\text{pad}} \in \mathbb{R}^{d \times k}$. As shown in Figure 1.A and D, red nodes are padding nodes. For the downsampling, we pad the input sequence on the boundary by adding nodes with the same node position and node features as the boundary node. For example, in Figure 1.A, the red node is the duplicate of the last white node. For the upsampling, we need both boundary padding and internal padding, similar to image padding in transpose convolution. The boundary padding is the same as that of downsampling. For an internal padding node, such as the second red node in Figure 1.D, it is initialized with the average value of the position and node features of its two nearest nodes on both sides.

**Edge Building.** After sequence padding, we perform an edge-building step to construct a graph from a padded protein sequence $\mathcal{S}_{\text{pad}}$. We could adopt fully connected graphs in order to capture interactions between all atom pairs. As shown in Figure 1.B, the edges in the constructed complete graph are in red. For simplicity, we only show the edge connections for one node. Note that ways of edge connections can be flexible in this step. Empirically we find that constructing a complete graph only over the non-padded sequence during downsampling gives better reconstruction performance.

**Graph Expansion.** Then, for the graph expansion step, we need to first initialize downsampled nodes and connect them to the graph constructed in the edge-building step. We denote the expanded graph as $\mathcal{G}_{\text{exp}} = (\boldsymbol{X}_{\text{exp}}, \boldsymbol{H}_{\text{exp}}, \boldsymbol{A}_{\text{exp}})$, where $\boldsymbol{X}_{\text{exp}} = [\boldsymbol{X}_{\text{pad}}, \boldsymbol{X}_{\text{down}}] \in \mathbb{R}^{3 \times (k + \frac{m}{2})}$, $\boldsymbol{H}_{\text{exp}} = [\boldsymbol{H}_{\text{pad}}, \boldsymbol{H}_{\text{down}}] \in \mathbb{R}^{d \times (k + \frac{m}{2})}$, and $\boldsymbol{A}_{\text{exp}} \in \mathbb{R}^{(k + \frac{m}{2}) \times (k + \frac{m}{2})}$. Specifically, we create a set of new nodes with positions $\boldsymbol{X}_{\text{down}} \in \mathbb{R}^{3 \times \frac{m}{2}}$ and node feature vectors $\boldsymbol{H}_{\text{down}} \in \mathbb{R}^{d \times \frac{m}{2}}$ which represent the downsampled sequence. The edge connections between downsampled sequence and the augmented protein sequence are created in a 1D CNN convention. Specifically, only nodes within a kernel-sized window will be connected to a new node. For example, as shown in Figure 1.B, the green area denotes a kernel of size 3, and the first black node connects to the first three white nodes in the green area. And each new node is initialized as the average of its connected nodes for both position and node feature.

**Equivariant Message Passing.** Next, we use an E(n) equivariant graph neural network (EGNN) (Satorras et al., 2021) to perform message passing on the expanded graph $\mathcal{G}_{\text{exp}}$ to update downsample nodes. Formally,

$$\hat{\boldsymbol{X}}_{\text{exp}}, \hat{\boldsymbol{H}}_{\text{exp}} = \text{EGNN}[\boldsymbol{X}_{\text{exp}}, \boldsymbol{H}_{\text{exp}}], \tag{4}$$

where $\hat{\boldsymbol{X}}_{\text{exp}} = [\hat{\boldsymbol{X}}, \hat{\boldsymbol{X}}_{\text{down}}]$ and $\hat{\boldsymbol{H}}_{\text{exp}} = [\hat{\boldsymbol{H}}, \hat{\boldsymbol{H}}_{\text{down}}]$. EGNN contains $L$ equivariant convolution layers (EGCL). Each layer performs a position and feature update, such that $\boldsymbol{x}_i^{l+1}, \boldsymbol{h}_i^{l+1} = \text{EGCL}[\boldsymbol{x}_i^l, \boldsymbol{h}_i^l]$, which is defined below:

$$\boldsymbol{m}_{ij} = \phi_e(\boldsymbol{h}_i^l, \boldsymbol{h}_j^l, d_{ij}^2, a_{ij}), \tag{5}$$

$$\boldsymbol{h}_i^{l+1} = \phi_h(\boldsymbol{h}_i^l, \sum_{j \neq i} \tilde{e}_{ij} \boldsymbol{m}_{ij}), \tag{6}$$

$$\boldsymbol{x}_i^{l+1} = \boldsymbol{x}_i^l + \sum_{j \neq i} \frac{\boldsymbol{x}_i^l - \boldsymbol{x}_j^l}{d_{ij} + 1} \phi_x(\boldsymbol{m}_{ij}), \tag{7}$$

where $d_{ij} = \left\| \boldsymbol{x}_i^l - \boldsymbol{x}_j^l \right\|_2$ denotes the Euclidean distance between nodes $i$ and $j$, and $a_{ij} = \text{MLP}([\boldsymbol{h}_i^l, \boldsymbol{h}_j^l])$ is the edge feature for edge $(i, j)$. Following Hoogeboom et al. (2022), we use $d_{ij} + 1$ to normalize the node distance to improve numerical stability and use an attention mechanism $\tilde{e}_{ij} = \phi_{inf}(\boldsymbol{m}_{ij})$ to infer a soft estimation of edges.

Then after the message passing, we will only keep the updated downsampled sequence $(\hat{\boldsymbol{X}}_{\text{down}}, \hat{\boldsymbol{H}}_{\text{down}})$ as the input of next layer, as shown in Figure 1.C. During the upsampling stage in the decoder, we perform the same four steps as introduced above. After upsampling to the original size of the input augmented protein, we obtain a reconstructed sequence with position and node embedding for each node. Then we use an MLP to process the final node embedding and predict whether a reconstructed node belongs to the augmented node type. We then use another MLP to predict the amino acid type of each node.

**Training Loss.** Reconstruction loss of autoencoder consists of six parts. First, we have a cross-entropy loss $\mathcal{L}_{\text{aug}}$ on a binary classification task to determine whether each reconstructed node is an augmented node that not belongs to the original protein. Next, we use another cross-entropy loss $\mathcal{L}_{\text{aa}}$ on the amino acid type prediction for each node. And then, we calculate the mean absolute error (MAE) of the position for each non-augmented node between the reconstructed protein and ground truth, and we denote it as $\mathcal{L}_{\text{pos}}$. Apart from these three losses, to further consider the secondary structure reconstruction for proteins, we also include edge distance loss $\mathcal{L}_{\text{dist}}$ and torsion angle loss $\mathcal{L}_{\text{tor}}$ calculated across the non-augmented nodes. Specifically, edge distance is calculated as the Euclidean distance between every two consecutive $C_\alpha$ atoms, and the torsion angle is the angle between two planes formed by four consecutive $C_\alpha$ atoms. To avoid latent node embeddings having an arbitrarily high variance, we use slight KL divergence loss $\mathcal{L}_{\text{reg}}$ to regularize latent node embeddings, which is similar to a variational autoencoder. So the total loss is the weighted sum of these individual losses. Formally,

$$\mathcal{L}_{\text{total}} = \mathcal{L}_{\text{aug}} + \mathcal{L}_{\text{aa}} + \mathcal{L}_{\text{pos}} + w_1 * \mathcal{L}_{\text{dist}} + w_2 * \mathcal{L}_{\text{tor}} + w_3 * \mathcal{L}_{\text{reg}}, \tag{8}$$

where $w_1$, $w_2$, and $w_3$ are relative weights to control the edge distance loss, torsion angle loss, and regularization loss, respectively. We want the network to optimize the absolute position of each node first and adjust edge distance and torsion angle later, so we set $w_1$ and $w_2$ as 0.5. Also, we want the autoencoder to have good reconstruction performance, so we only use very small regularization, and we set $w_3$ equal to $1e^{-4}$.

### 3.3 Latent Diffusion

Modeling the extracted latent representations $(\boldsymbol{X}_{\text{down}}, \boldsymbol{H}_{\text{down}})$ of protein backbone structures poses unique challenges due to the fact that they consist of 3D Euclidean positions, which differ from images and texts. In this section, we first explain the desired distribution E(3) invariance property and then provide a detailed description of the latent diffusion process that satisfies this property for the task of protein backbone generation. In this section, $p_{\text{data}}$, $p_{\text{model}}$, and $p_\theta$ denote the underlying data distribution, the output distribution of the whole model framework, and the latent distribution from the latent diffusion model, respectively.

**Distribution E(3) Invariance**. For a given protein backbone structure $(\boldsymbol{X}, \boldsymbol{H})$, we would like the learned data distribution to be E(3) invariant: $p_{\text{data}}(\boldsymbol{X}, \boldsymbol{H}) = p_{\text{data}}(R\boldsymbol{X} + b, \boldsymbol{H})$ as the geometric 3D structure remains unchanged after E(3) transformations, where $R \in \mathbb{R}^{3 \times 3}$, $|R| = \pm 1$ describing the rotation and reflection transformations and $b \in \mathbb{R}^3$ for translation in 3D space. Because our protein autoencoder is translation invariant as described in Sec. 3.2, $p_{\text{model}}(\boldsymbol{X}, \boldsymbol{H}) = p_{\text{model}}(\boldsymbol{X} + b, \boldsymbol{H})$ holds naturally. Hence, distribution rotation and reflection invariance $p_{\text{model}}(\boldsymbol{X}, \boldsymbol{H}) = p_{\text{model}}(R\boldsymbol{X}, \boldsymbol{H})$ needs to be satisfied for the latent diffusion process.

In our approach, we propose to decompose the generation of protein backbone structures into two stages, including (1) protein latent representation generation and (2) latent representation decoding. The model distribution can be defined as $p_{\text{model}}(\boldsymbol{X}, \boldsymbol{H}) = f_{\text{decoder}}(p_\theta(\boldsymbol{X}_{\text{down}}, \boldsymbol{H}_{\text{down}}))$. Given that the decoding process is E(3) equivariant and deterministic, if the latent diffusion model $\boldsymbol{s}_\theta$ satisfies $p_\theta(\boldsymbol{X}_{\text{down}}, \boldsymbol{H}_{\text{down}}) = p_\theta(R\boldsymbol{X}_{\text{down}} + b, \boldsymbol{H}_{\text{down}})$, the distribution rotation and reflection invariance $p_{\text{model}}(\boldsymbol{X}, \boldsymbol{H}) = p_{\text{model}}(R\boldsymbol{X} + b, \boldsymbol{H})$ can be satisfied.

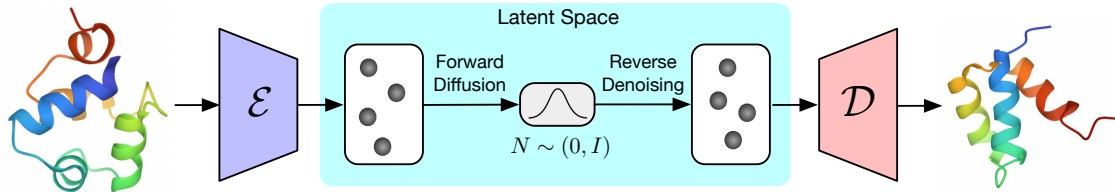

Figure 2: Pipeline of LatentDiff. Encoder $\mathcal{E}$ and decoder $\mathcal{D}$ are pretrained via equivariant protein autoencoder introduced in Section 3.2, and their parameters are fixed during training the latent diffusion. Protein structures are encoded into latent representations via the encoder $\mathcal{E}$. And latent representations are gradually perturbed into Gaussian noise. During generation, we first sample Gaussian noise and use the learned denoising network to generate protein representations in the latent space. And then, the decoder $\mathcal{D}$ decodes latent representations to protein structures.

The challenge of $p_\theta(\boldsymbol{X}_{\mathrm{down}}, \boldsymbol{H}_{\mathrm{down}}) = p_\theta(R\boldsymbol{X}_{\mathrm{down}} + b, \boldsymbol{H}_{\mathrm{down}})$ can be addressed by (1) modeling zero-mean geometric distribution for $\boldsymbol{X}$, (2) using a high-dimensional Gaussian distribution as the prior distribution, and (3) employing rotation and reflection equivariant reverse diffusion process (Hoogeboom et al., 2022; Xu et al., 2021). Specifically, the influence of translation transformations in 3D space is omitted by reducing the central position of $\boldsymbol{X}$. Additionally, by using an isotropic high dimensional Gaussian prior, we have $p_\theta(\boldsymbol{X}_T, \boldsymbol{H}_T) = p_\theta(R\boldsymbol{X}_T, \boldsymbol{H}_T)$. The rotation and reflection equivariant reverse diffusion process further guarantees that $p_\theta(\boldsymbol{X}_t, \boldsymbol{H}_t) = p_\theta(R\boldsymbol{X}_t, \boldsymbol{H}_t)$ for any time $t$ and the proof is provided in Appendix. A.1.

**Rotation and Reflection Invariant Latent Diffusion.** Due to the aforementioned considerations, we propose the rotation and reflection distribution invariant latent forward and reverse diffusion processes for the extracted protein backbone latent features $(\boldsymbol{X}_{\mathrm{down}}, \boldsymbol{H}_{\mathrm{down}})$. The implementation is based on Hoogeboom et al. (2022) with adjustments to support the latent diffusion process. The pipeline of our protein latent diffusion is shown in Figure 2. During the forward process, the input latent representations $(\boldsymbol{X}_{\mathrm{down}}, \boldsymbol{H}_{\mathrm{down}})$ are diffused slowly into random noise by a sequence of noise scales $0 < \beta_1, \beta_2, \ldots, \beta_N < 1$ as follows

$$\boldsymbol{X}_i = \sqrt{1 - \beta_i}\boldsymbol{X}_{i-1} + \sqrt{\beta_i}\boldsymbol{\sigma_X},$$
$$\boldsymbol{H}_i = \sqrt{1 - \beta_i}\boldsymbol{H}_{i-1} + \sqrt{\beta_i}\boldsymbol{\sigma_H},$$

where $\boldsymbol{\sigma_H} \sim \mathcal{N}(\boldsymbol{0}, \mathbf{I})$, and $\boldsymbol{\sigma_X}$ is first sampled from $\mathcal{N}(\boldsymbol{0}, \mathbf{I})$ and then reduced based on the corresponding central position following Hoogeboom et al. (2022). And the closed-form forward process can be written as

$$\boldsymbol{X}_t = \sqrt{\alpha_t}\boldsymbol{X}_{\mathrm{down}} + \sqrt{1 - \alpha_t}\boldsymbol{\sigma_X}, \tag{9}$$
$$\boldsymbol{H}_t = \sqrt{\alpha_t}\boldsymbol{H}_{\mathrm{down}} + \sqrt{1 - \alpha_t}\boldsymbol{\sigma_H}, \tag{10}$$

where $\alpha_t = \prod_{i=0}^{t}(1 - \beta_i)$. Since $\alpha_t$ is a scalar value, we have $p_t(\boldsymbol{X}_t, \boldsymbol{H}_t) = p(\boldsymbol{X}_{\mathrm{down}}, \boldsymbol{H}_{\mathrm{down}})p(\boldsymbol{\sigma_X}, \boldsymbol{\sigma_H})$ where $p_t$ is the data distribution at time $t$ and $p(\boldsymbol{\sigma_X}, \boldsymbol{\sigma_H}) = p(\boldsymbol{\sigma_X})p(\boldsymbol{\sigma_H})$ denotes the corresponding multivariate Gaussian distributions. It can be seen that $p_t(\boldsymbol{X}_t, \boldsymbol{H}_t) = p_t(R\boldsymbol{X}_t, \boldsymbol{H}_t)$ because $p(\boldsymbol{X}_{\mathrm{down}}, \boldsymbol{H}_{\mathrm{down}})p(\boldsymbol{\sigma_X}, \boldsymbol{\sigma_H}) = p(R\boldsymbol{X}_{\mathrm{down}}, \boldsymbol{H}_{\mathrm{down}})p(\boldsymbol{\sigma_X}, \boldsymbol{\sigma_H})$. Hence, the forward diffusion process satisfies rotation and reflection distribution invariance.

For the reverse diffusion process, a reverse Markov chain is formed as below

$$(\boldsymbol{X}_{t-1}, \boldsymbol{H}_{t-1}) = \frac{1}{\sqrt{1 - \beta_t}}\boldsymbol{\mu_t} + \sqrt{\beta_t}(\boldsymbol{\sigma_X}, \boldsymbol{\sigma_H}),$$
$$\boldsymbol{\mu_t} = (\boldsymbol{X}_t, \boldsymbol{H}_t) - \frac{\beta_t}{\sqrt{1 - \alpha_t}}\boldsymbol{s_\theta}(\boldsymbol{X}_t, \boldsymbol{H}_t, t),$$

where $\boldsymbol{s_\theta}$ is a rotation and reflection equivariant network implemented based on EGNN (Satorras et al., 2021).

**Training Loss**. The reverse diffusion model $s_\theta$ is trained with a re-weighted evidence lower bound (ELBO) following ProtDiff (Trippe et al., 2022) and DDPM (Ho et al., 2020) as below

$$\theta^\star = \text{argmin}_\theta \mathbb{E}_{t,(\boldsymbol{X}_{\text{down}}, \boldsymbol{H}_{\text{down}}),\boldsymbol{\sigma}}[\|\boldsymbol{\delta}\|^2], \tag{11}$$

$$\boldsymbol{\delta} = \boldsymbol{\sigma} - s_\theta(\sqrt{\alpha_t}(\boldsymbol{X}_{\text{down}}, \boldsymbol{H}_{\text{down}}) + \sqrt{1 - \alpha_t}\boldsymbol{\sigma}, t), \tag{12}$$

where $\boldsymbol{\sigma} = (\boldsymbol{\sigma_X}, \boldsymbol{\sigma_H})$.

### 3.4 Overall Generation Process

We have introduced the main components of our protein latent diffusion model, LatentDiff. To generate a novel protein backbone structure, we first sample multivariate Gaussian noise and use the learned latent diffusion model to generate 3D positions and node embeddings in the latent space. To further improve generation quality, we also use low-temperature sampling (Ingraham et al., 2022) to guide the reverse process in the diffusion model. And then we use the pre-trained decoder to generate backbone structures in the protein space. Note that the output of the decoder has a pre-defined fixed size. In order to generate proteins of various lengths, each node in the decoder output is predicted to be an augmented node or not. We simply find the first node that is classified as an augmented node and drop the remaining nodes in the generated protein backbone structure. Note that we do not use reconstructed amino acid types for the corresponding node. Instead, we use the inverse folding model ProteinMPNN (Dauparas et al., 2022) to predict protein amino acid sequences from generated backbone structures.

## 4 Experiments

We empirically demonstrate the effectiveness and efficiency of our method for generating protein backbone structures. In Section 4.1, we first introduce the dataset we curated from existing protein databases and the benchmarking baseline models. In Section 4.2–Section 4.7, we show the reconstruction performance of the pre-trained autoencoder, the quality and diversity of generated protein backbone structures, and the parallel sampling efficiency of LatentDiff. In Appendix A.3, we describe the training details of the autoencoder and latent diffusion model.

### 4.1 Experimental Setting

**Dataset.** We curate the dataset from Protein Data Bank (PDB) and Swiss-Prot data in AlphaFold Protein Structure Database (AlphaFold DB) (Jumper et al., 2021; Varadi et al., 2022). Details of the dataset can be found in Appendix A.2. Note that the dataset employed in our research is larger than the ones used in both ProtDiff (Trippe et al., 2022) and FoldingDiff (Wu et al., 2022a). Our methodology necessitates that the latent space is well-structured and equivalent to the protein space. To realize this goal to the maximum extent, it is imperative that we utilize the maximum possible volume of data to train our model.

**Baselines.** To evaluate our proposed methods, we compare with the state-of-the-art protein backbone structure generation methods including ProtDiff (Trippe et al., 2022) and FoldingDiff (Wu et al., 2022a). Both of them are proposed to generate novel protein backbone level structures as discussed in Sec. 2.3.

### 4.2 Autoencoder Reconstruction

In this section, we demonstrate the reconstruction performance of the protein autoencoder. We compare autoencoders with different downsampling factors $f = \{2, 4, 8\}$, which we denote as $auto - f$.

**Metrics.** First, we evaluate the classification accuracy of augmented and non-augmented nodes (Augment Acc), and the accuracy of amino acid type classification (Residue Acc). And we have the following three geometric evaluations. We use root mean square deviation (RMSD) to compare the absolute position error between reconstructed $C_\alpha$ atoms and ground truth. Additionally, we measure edge stability, which counts the proportion of $C_\alpha - C_\alpha$ distance that resides with range [3.65Å, 3.95Å]. The reason for choosing this range is that 99% $C_\alpha - C_\alpha$ distances in ground truth are within this range. We also calculate the mean

Table 1: Performance of autoencoder with different downsampling factors. ↑ (↓) represents that a higher (lower) value indicates better performance.

| Factor | RMSD (Å)$^\downarrow$ | Augment Acc (%)$^\uparrow$ | Residue Acc (%)$^\uparrow$ | Edge Stable (%)$^\uparrow$ | Torsion MAE (rad)$^\downarrow$ |
|--------|------|-------------|-------------|-------------|--------------|
| 2 | 1.1607 | 100 | 99 | 68.61 | 1.1558 |
| 4 | 1.4070 | 100 | 98 | 61.66 | 1.2932 |
| 8 | 2.9488 | 100 | 48 | 52.54 | 1.8518 |

Table 2: Percentage of generated proteins with scTM score > 0.5. Following FoldingDiff and ProtDiff, results are shown within short (50–70) and long (70–128) categories.

| Method | 50–70 | 70–128 | 50–128 |
|--------|-------|--------|--------|
| ProtDiff | 17.1% | 8.9% | 11.8% |
| FoldingDiff | 27.1% | 9.4% | 14.2% |
| LatentDiff | 31.1% | 35.6% | 31.6% |

absolute error (MAE) of the torsion angle. Note that all the geometric evaluations are performed on the original protein backbones without considering augmented nodes.

In Table 1, we summarize the results with respect to these five metrics for protein autoencoders with different downsampling factors. In order to reduce the modeling space of proteins and make it easier for the diffusion model to learn the latent distribution, larger downsampling factors are preferred; but meanwhile, it will become more difficult to achieve good reconstruction results. We can see that $auto-8$ has the worst reconstruction performance because the autoencoder compresses information too much. Although $auto-2$ performs the best among the three settings, the number of nodes in the latent space is still relatively large. So in order to achieve a balance between computation and reconstruction performance, we finally choose $auto-4$ as the pre-trained model for generating latent space data and decoding protein backbones.

### 4.3 *In-silico* Evaluation

After sampling protein backbone structures, we also need to evaluate the designability of these generated structures. This means that for a generated backbone, whether we can connect some amino acids into a sequence and the sequence can naturally fold into that desired backbone structure. The most faithful and desirable evaluation is to check through a wet-lab experiment, but this is often resource demanding and not feasible. Here we use *in silico* evaluations as an alternative.

Specifically, for a generated backbone structure, we first use an inverse folding model, ProteinMPNN (Dauparas et al., 2022), to predict eight amino acid sequences that could possibly fold into that backbone structure. OmegaFold (Wu et al., 2022b) is then used to predict folding structures for each amino acid sequence. Next, we adopt TMalign (Zhang & Skolnick, 2005) to compute the similarity between the generated backbone structure and each OmegaFold-predicted backbone structure and calculate a TM score to quantify the similarity. The maximum TM-score among these eight scores is referred to as the self-consistency TM-score (scTM). If a scTM score is larger than 0.5, two backbone structures are considered with the same fold and that generated backbone structure is designable. Similar as in previous works (Wu et al., 2022a; Trippe et al., 2022), we generate 780 backbone structures with various lengths between 50 and 128 and evaluate them by the scTM score, for which the sampling temperature in ProteinMPNN is 0.1. The histogram of scTM scores is shown in Figure 3, and the comparison with FoldingDiff and ProtDiff is shown in Table 2. For our LatentDiff, 247 of 780 (31.6%) generated structures have their scTM scores > 0.5. The achieved percentage of designable structures has a significant margin over FoldingDiff (14.2%) and ProtDiff (11.8%). In the reported results by these two baseline methods, the authors grouped their generated structures into short (50-70) and long (70-128) ones, and reported the designable percentage (scTM > 0.5) within each category. For the short category, 31.1% of our generated structures are designable, which is higher than FoldingDiff (27.1%) and ProtDiff (17.1%). For the long category, our percentage (35.6%) is in fact much better than these two baseline models (8.9% for ProtDiff and 9.4% for FoldingDiff), demonstrating significantly improved scalability

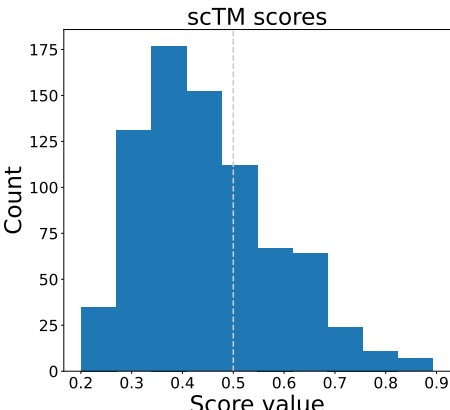

Figure 3: scTM score distribution of generated backbone structures with length between 50 and 128. 31.6% (247/780) generated samples are designable (scTM > 0.5)

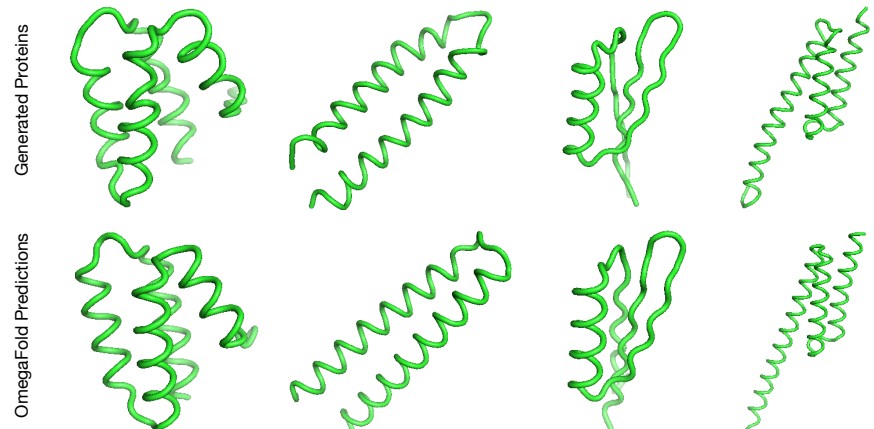

Figure 4: Some samples of generated structures with scTM > 0.5. The top row shows our generated backbones and the second row shows the backbone structures predicted by the OmegaFold from the predicted amino acid sequences. We use the inverse folding model ProteinMPNN to generate these amino acid sequences that can likely to fold into our generated structures.

due to our designed model space reduction in LatentDiff. We also visualize some exemplar backbones and OmegaFold-predicted backbone structures using PyMOL (DeLano, 2002) in Figure 4.

### 4.4 Structure Distribution Analysis

After showing the success of *in silico* tests, we illustrate the distributions of generated samples in both the original protein space and the latent space. First, we show the edge distance and bond angle distributions of generated backbones and test set backbones. As shown in Figure 5, the distributions of generated samples are similar to the test distributions. We further investigate the distributions in the latent space. Specifically, we show the distributions of node positions, edge distances, and node embeddings in the latent space. For simplicity, we only show the $x$ coordinate of the latent node position and the first dimension of latent node embeddings. As shown in Figure 6, these distributions of generated latent samples almost recover the latent training data distributions.

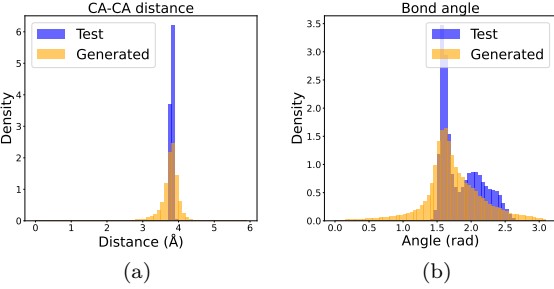

Figure 5: Distribution comparison between generated backbone structures and test set protein backbones. (a) Edge distance between any two consecutive $C_\alpha$ atoms along a protein chain. (b) Bond angle formed by any three consecutive $C_\alpha$ atoms along a protein chain.

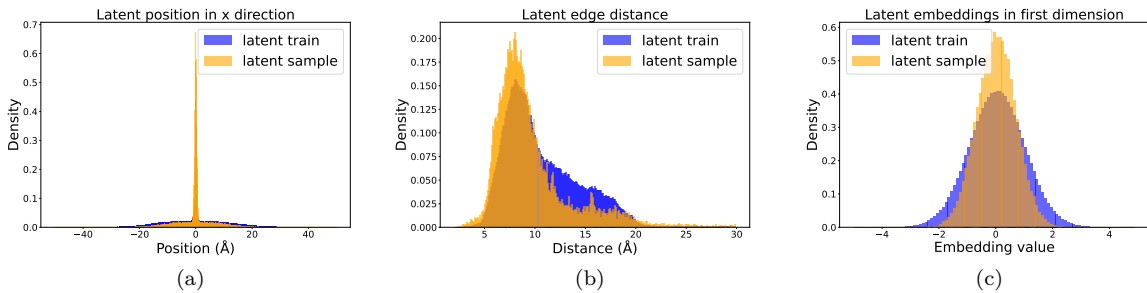

Figure 6: Distribution comparison between training data and generated samples in the latent space. (a) Position of latent node in the $x$ direction. (b) Edge distance between any two consecutive nodes in the latent space. (c) First dimension of latent node embeddings.

### 4.5 Secondary Structures

We use P-SEA (Labesse et al., 1997) to count the number of two types of secondary structures in the generated proteins with scTM > 0.5. Specifically, we calculate the percentage of generated proteins that contain only $\alpha$-helix, only $\beta$-sheet, and both $\alpha$-helix and $\beta$-sheet, respectively. The results are shown in Table 3. As seen, more than half of the generated proteins include $\alpha$-helix, and a large portion of generated proteins contain $\beta$-sheet. This proves that our method can generate various secondary structures in natural proteins.

### 4.6 Diversity

We also evaluate the diversity of generated proteins with scTM > 0.5 (designable), as shown in Table 4. Specifically, we calculate the TM scores with all other designable proteins for each designable protein and choose the maximum TM score to measure its similarity with the generated proteins. Then, we calculate the average of maximum TM scores over all designable proteins to assess the diversity of the generated proteins (lower is better). From the table, we can see that LatentDiff can generate more diverse protein structures than ProtDiff and is comparable with FoldingDiff.

### 4.7 Parallel Sampling Efficiency Comparison

In this section, we demonstrate the parallel sampling efficiency of our method. Diffusion models usually need to perform thousands of reverse steps to generate a single data point, and the data size must be the same during every reverse step. So this generation process is very time-consuming and computationally expensive,

Table 3: Percentage of generated proteins that contain only $\alpha$-helix, only $\beta$-sheet, and both $\alpha$-helix and $\beta$-sheet, respectively.

| $\alpha$-HELIX ONLY | $\beta$-SHEET ONLY | $\alpha$-HELIX + $\beta$-SHEET |
|---|---|---|
| 58.7% | 13.3% | 14.9% |

Table 4: Diversity of generated designable proteins (scTM > 0.5). ↓ represents that a lower value indicates better performance.

| METHOD | DIVERSITY$^{\downarrow}$ |
|---|---|
| PROTDIFF | 0.836±0.1648 |
| FOLDINGDIFF | 0.585±0.1276 |
| LATENTDIFF | 0.615±0.0849 |

Table 5: Sampling efficiency comparison between diffusion models in latent and protein space. LatentDiff-P denotes no protein autoencoder being used and diffusion is performed directly in the protein space.

| METHOD | PARAMETERS | PROTEIN LENGTH | LATENT NODE | DIFFUSION STEPS | TIME (HRS) | SPEED (SEC/SAMPLE) |
|---|---|---|---|---|---|---|
| PROTDIFF | 1974528 | 128 | N/A | 1000 | 1.9 | 6.85 |
| LATENTDIFF-P | 2016453 | 128 | N/A | 1000 | 2.9 | 10.66 |
| LATENTDIFF | 2027984 | 128 | 32 | 1000 | 0.18 | 0.68 |
| LATENTDIFF | 2027984 | 128 | 32 | 2000 | 0.36 | 1.33 |
| LATENTDIFF | 2027984 | 256 | 64 | 1000 | 0.73 | 2.66 |

especially when the modeling space of diffusion models is large. So this prohibits efficient parallel sampling with limited computing resources.

Generation in latent space can reduce memory usage and computational complexity as the latent space is much smaller than the protein space, thereby improving the generation throughput. The reason we compare efficiency in terms of parallel sampling is that, in practice, it requires sampling a large amount of proteins in the screening procedure, so high throughput sampling is desired. In this sense, sampling in latent space demonstrates significant efficiency improvement. For the experiments, we compare sampling 1000 proteins with different methods on a single NVIDIA 2080Ti GPU and summarize the result in Table 5. Note that these experiments are only used to test sampling efficiency, and the network weights are just randomly initialized. For fair comparison and to rule out the other factors other than different modeling space, we compare with ProtDiff and our LatentDiff without downsampling (named LatentDiff-P), as denoising networks for these models are similar, and we also make the number of parameters to be similar for these models. For our model, the processing time of the decoder is orders of magnitude less than that of our latent diffusion model, so we do not take the decoder time into account. From the result, we can see that the generation time of 1000 protein structures in the protein space is about 2.9 hours, while it only takes about 11 minutes to generate in the latent space and then map to the protein space. So reducing modeling space demonstrates potential usefulness in practice. The sampling time of LatentDiff scales linearly with the number of diffusion steps because diffusion steps are performed sequentially. Moreover, since we use a fully connected graph for the diffusion model, increasing latent nodes will quadratically increase memory consumption and computational complexity. Consequently, the sampling throughput will decrease and is contingent upon the GPU memory and computational capacity, with the throughput being constrained by whichever resource reaches its limit first.

## 5    Discussion

In this section, we discuss the limitations of our LatentDiff and potential future directions beyond LatentDiff.

**Limitations**. For one thing, the input protein backbone structure needs to be padded to a fixed length for the protein autoencoder, and the actual length is predicted during the decoding process. For another thing,

due to the modeling difficulty of structure and sequence co-design mentioned by previous works (Trippe et al., 2022; Wu et al., 2022a), we only use the generated protein backbone structures but corresponding generated sequences are not used. We think the reason for co-design not working well is that directly modeling the joint distribution of structures and sequences is very hard since the model needs to learn the complex correlation between protein structures and amino acid sequences. This somewhat requires the generative model to implicitly learn the inverse folding or folding process, which by themselves are complex tasks that need powerful inverse folding or protein folding prediction models to solve. However, using the high-accuracy inverse folding model to predict sequences from structures can reduce the burden of the generative model to learn the complex correlation between protein structures and amino acid sequences.

**Future Directions**. There are several potential directions beyond the current LatentDiff and we leave them to future works. (1) The 3D protein autoencoder can be adjusted to support arbitrary length input and generate arbitrary length protein backbone structure. (2) Due to limited computation resources, we cannot train our 3D protein autoencoder on all protein structures predicted by AlphaFold (Varadi et al., 2022). And the performance of the 3D protein autoencoder is promising to boost when more training protein structures are available. (3) The length of generated proteins is limited to 128 in our work. With more data and computing resources, our method has the potential capability to generate longer proteins and then can generate proteins that exhibit more diverse folds. (4) There's still an opportunity to improve structure and sequence co-design. With a more powerful protein autoencoder obtained by (1) and (2), the modeling difficulty of structure and sequence co-design may be addressed naturally by our proposed LatentDiff framework. Besides, an iterative refinement approach involving alternating between sequence and structure generation steps might also be useful to gradually improve the consistency between the generated sequence and structure. In addition, incorporating physical constraints during the generative modeling process, such as integrating physical principles such as energy functions and geometric constraints into the generative model, could also be possible to guide the generation process to produce sequences that are more likely to fold into the generated structures. (5) Conditional generation tasks are very useful in practice as they enable protein generation with desired properties and are worth more exploration in future work.

# 6 Broader Impact Statement

Our protein generation method, enabling the production of novel proteins, has significant broader impact potential. On one hand, it might offer potential opportunities for advancements in medicine, agriculture, and biotechnology, facilitating the development of innovative therapeutics, enzymes, and biomaterials in the future. On the other hand, while considering the concerns raised regarding the computational selection of potentially dangerous agents, we should prioritize responsible research practices, with stringent safety protocols, adherence to regulations, and collaboration with biosecurity experts to ensure the responsible handling of generated proteins. By fostering collaboration and knowledge dissemination, we aim to advance protein design while actively managing any potential risks associated with our method.

# 7 Conclusion

We have proposed LatentDiff, a 3D latent diffusion framework for protein backbone structure generation. To reduce the modeling space of protein structures, LatentDiff uses a pre-trained equivariant 3D autoencoder to transform protein backbones into a more compact latent space, and models the latent distribution with an equivariant latent diffusion model. LatentDiff is shown to be effective and efficient in generating designable protein backbone structures by comprehensive experimental results.

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

# A    Appendix

## A.1    Distribution Rotation and Reflection Invariant Reverse Diffusion Process

In this section, we provide proof that by (1) using a high-dimensional Gaussian distribution as the prior distribution, and (2) employing rotation and reflection equivariant reverse diffusion model $s_{\boldsymbol{\theta}}$ Hoogeboom et al. (2022); Xu et al. (2021), the challenge of $p_{\theta}(\boldsymbol{X}_{\text{down}}, \boldsymbol{H}_{\text{down}}) = p_{\theta}(R\boldsymbol{X}_{\text{down}}, \boldsymbol{H}_{\text{down}})$ can be addressed. The proof process borrows ideas from Xu et al. (2021) and Hoogeboom et al. (2022).

First, because $p_{\theta}(\boldsymbol{X}_T, \boldsymbol{H}_T) = \mathcal{N}(\boldsymbol{0}, \boldsymbol{I})$, and $\mathcal{N}(\boldsymbol{0}, \boldsymbol{I})$ is isotropic, we have $p_{\theta}(\boldsymbol{X}_T, \boldsymbol{H}_T) = p_{\theta}(R\boldsymbol{X}_T, \boldsymbol{H}_T)$, where $R \in \mathbb{R}^{3\times3}$, $|R| = \pm 1$ describes the rotation and reflection transformations in 3D space.

Second, because $s_{\boldsymbol{\theta}}$ is rotation and reflection equivariant for $\boldsymbol{X}_t$ and rotation and reflection invariant for $\boldsymbol{H}_t$, and

$$\boldsymbol{X}_{t-1} = \frac{1}{\sqrt{1-\beta_t}}(\boldsymbol{X}_t - \frac{\beta_t}{\sqrt{1-\alpha_t}}s_{\boldsymbol{\theta}}(\boldsymbol{X}_t, \boldsymbol{H}_t, t)_{\boldsymbol{X}}) + \sqrt{\beta_t}\boldsymbol{\sigma_X}, \tag{13}$$

$$\boldsymbol{H}_{t-1} = \frac{1}{\sqrt{1-\beta_t}}(\boldsymbol{H}_t - \frac{\beta_t}{\sqrt{1-\alpha_t}}s_{\boldsymbol{\theta}}(\boldsymbol{X}_t, \boldsymbol{H}_t, t)_{\boldsymbol{H}}) + \sqrt{\beta_t}\boldsymbol{\sigma_H}, \tag{14}$$

where $s_{\boldsymbol{\theta}}(\boldsymbol{X}_t, \boldsymbol{H}_t, t)_{\boldsymbol{X}}$ and $s_{\boldsymbol{\theta}}(\boldsymbol{X}_t, \boldsymbol{H}_t, t)_{\boldsymbol{H}}$ denote the network predictions to update $\boldsymbol{X}$ and $\boldsymbol{H}$, correspondingly. When we apply transformation $R \in \mathbb{R}^{3\times3}$, $|R| = \pm 1$ to $\boldsymbol{X}_{t-1}$, we will have

$$R\boldsymbol{X}_{t-1} = \frac{1}{\sqrt{1-\beta_t}}R(\boldsymbol{X}_t - \frac{\beta_t}{\sqrt{1-\alpha_t}}s_{\boldsymbol{\theta}}(\boldsymbol{X}_t, \boldsymbol{H}_t, t)_{\boldsymbol{X}}) + \sqrt{\beta_t}R\boldsymbol{\sigma_X} \tag{15}$$

$$= \frac{1}{\sqrt{1-\beta_t}}(R\boldsymbol{X}_t - \frac{\beta_t}{\sqrt{1-\alpha_t}}Rs_{\boldsymbol{\theta}}(\boldsymbol{X}_t, \boldsymbol{H}_t, t)_{\boldsymbol{X}}) + \sqrt{\beta_t}R\boldsymbol{\sigma_X} \tag{16}$$

$$= \frac{1}{\sqrt{1-\beta_t}}(R\boldsymbol{X}_t - \frac{\beta_t}{\sqrt{1-\alpha_t}}s_{\boldsymbol{\theta}}(R\boldsymbol{X}_t, \boldsymbol{H}_t, t)_{\boldsymbol{X}}) + \sqrt{\beta_t}R\boldsymbol{\sigma_X}, \tag{17}$$

and we can have the following

$$p_{\theta}(\boldsymbol{X}_{t-1}, \boldsymbol{H}_{t-1}|\boldsymbol{X}_t, \boldsymbol{H}_t) = p_{\theta}(\boldsymbol{X}_t, \boldsymbol{H}_t)p(\boldsymbol{\sigma_X}, \boldsymbol{\sigma_H}) = p_{\theta}(R\boldsymbol{X}_t, \boldsymbol{H}_t)p(R\boldsymbol{\sigma_X}, \boldsymbol{\sigma_H}) = p_{\theta}(R\boldsymbol{X}_{t-1}, \boldsymbol{H}_{t-1}|R\boldsymbol{X}_t, \boldsymbol{H}_t). \tag{18}$$

Beyond this, for the reverse diffusion time $t \in \{T, T-1, \cdots, 1\}$, assume $p_{\theta}(\boldsymbol{X}_t, \boldsymbol{H}_t)$ satisfies $p_{\theta}(\boldsymbol{X}_t, \boldsymbol{H}_t) = p_{\theta}(R\boldsymbol{X}_t, \boldsymbol{H}_t)$, where $R \in \mathbb{R}^{3\times3}$, $|R| = \pm 1$ describes the rotation and reflection transformations in 3D space. Then we have:

$$p_{\theta}(R\boldsymbol{X}_{t-1}, \boldsymbol{H}_{t-1}) = \int_{(\boldsymbol{X}_t, \boldsymbol{H}_t)} p_{\theta}(R\boldsymbol{X}_{t-1}, \boldsymbol{H}_{t-1}|\boldsymbol{X}_t, \boldsymbol{H}_t)p_{\theta}(\boldsymbol{X}_t, \boldsymbol{H}_t)$$

$$= \int_{(\boldsymbol{X}_t, \boldsymbol{H}_t)} p_{\theta}(R\boldsymbol{X}_{t-1}, \boldsymbol{H}_{t-1}|RR^{-1}\boldsymbol{X}_t, \boldsymbol{H}_t)p_{\theta}(RR^{-1}\boldsymbol{X}_t, \boldsymbol{H}_t)$$

$$= \int_{(\boldsymbol{X}_t, \boldsymbol{H}_t)} p_{\theta}(\boldsymbol{X}_{t-1}, \boldsymbol{H}_{t-1}|R^{-1}\boldsymbol{X}_t, \boldsymbol{H}_t)p_{\theta}(R^{-1}\boldsymbol{X}_t, \boldsymbol{H}_t),$$

let $\boldsymbol{X}' = R^{-1}\boldsymbol{X}_t$, we have $\det R = 1$ and

$$p_{\theta}(R\boldsymbol{X}_{t-1}, \boldsymbol{H}_{t-1}) == \int_{(\boldsymbol{X}', \boldsymbol{H}_t)} p_{\theta}(\boldsymbol{X}_{t-1}, \boldsymbol{H}_{t-1}|\boldsymbol{X}', \boldsymbol{H}_t)p_{\theta}(\boldsymbol{X}', \boldsymbol{H}_t) * \det R = p_{\theta}(\boldsymbol{X}_{t-1}, \boldsymbol{H}_{t-1}), \tag{19}$$

and $p_{\theta}(\boldsymbol{X}_{t-1}, \boldsymbol{H}_{t-1})$ is invariant. By induction, $p_{\theta}(\boldsymbol{X}_{T-1}, \boldsymbol{H}_{T-1}), \ldots, p_{\theta}(\boldsymbol{X}_0, \boldsymbol{H}_0)$ are all invariant and the proof is complete.

## A.2    Datasets

We curate the dataset from Protein Data Bank (PDB) and Swiss-Prot data in AlphaFold Protein Structure Database (AlphaFold DB) (Jumper et al., 2021; Varadi et al., 2022). We filter all the single-chain protein

data from PDB with $C_\alpha$ – $C_\alpha$ distance less than 5Å and sequence length between 40 and 128 residues, resulting in 4460 protein sequences. We randomly split the data according to 80/10/10 train/validation/test split. In order to include more training data, we further curate protein data from two resources and add them to the current training set. The first part of augmented training data comes AlphaFold DB. Specifically, we filter single-chain proteins in Swiss-Prot with lengths between 40 and 128 and add these proteins to the training data. The second part of augmented training data comes from PDB, where we curate data from those single-chain proteins with $C_\alpha$ – $C_\alpha$ distance larger than 5Å and sequence lengths longer than 40. Specifically, we split these proteins at the position where $C_\alpha$ – $C_\alpha$ distance is larger than 5Å to obtain protein fragments. Then we add these fragments with lengths between 50 and 128 to the training data. For these fragments with lengths longer than 256, we uniformly cut them into lengths between 50 and 128, and add them to the training data. After this data augmentation process, we can finally obtain about $100k$ training data.

### A.3 Experimental Details

For training of the autoencoder, we have used all the available training data. We then use the trained encoder to embed all the training protein data and use their latent representations to train the latent diffusion model. We have trained the autoencoder for 200 epochs with batch size 128, by Adam optimizer (Kingma & Ba, 2015) with learning rate $1e^{-3}$, $\beta_1 = 0.9$, $\beta_2 = 0.999$, and weight delay $2e^{-4}$. The latent diffusion model has been trained for $13M$ steps with batch size 128, by Amsgrad optimizer (Reddi et al., 2018) with learning rate $5e^{-5}$, $\beta_1 = 0.9$, $\beta_2 = 0.999$, and weight delay $1e^{-12}$. We use 1000 diffusion steps and the same noise scheduler used in Hoogeboom et al. (2022). We have implemented all the models in PyTorch and have trained all the models on a single NVIDIA A100 GPU.

### A.4 Latent Space Interpolation

Usually, it is natural to visualize the latent space and perform latent code interpolation to test if the latent space is well-structured. However, a protein in our latent space is not represented by a single latent feature vector, but rather, it is a set of nodes associated with 3D coordinates and node features. As such, it is difficult to use dimension reduction techniques like t-SNE to visualize the latent space. In addition, we did not add a KL-divergence loss on coordinates since it would break equivariance. Even for invariant node features, we only add a minimal KL-divergence penalty to control the variance of the latent space, as we aim to maintain high reconstruction accuracy for the autoencoder. Therefore, in our case, the latent space does not necessarily need to be well-structured, and arbitrary interpolation may not guarantee valid protein structures upon decoding.

To show this, we pick two generated proteins with scTM>0.5 (designable), and their corresponding latent space data are $(X_{emb}^s, H_{emb}^s)$ and $(X_{emb}^t, H_{emb}^t)$. Then we interpolate these two latent space data as $(X_{emb}^{interp}, H_{emb}^{interp}) = (X_{emb}^s * (1 - \lambda) + X_{emb}^t * \lambda, H_{emb}^s * (1 - \lambda) + H_{emb}^t * \lambda)$. We choose different values of $\lambda$ and decode the interpolated latent space data into proteins and calculate the scTM score, as shown in Table 6. We can see that if $\lambda$ is close to 0 or 1, generated proteins are still designable. However, if $\lambda$ is near 0.5, generated proteins are not valid, just as we analyzed above.

Table 6: The scTM score of proteins decoded from the interpolation of two latent protein representations. $\lambda$ is the interpolation weights. TM-left means the TM score with the start protein, and TM-right means the TM score with the end protein.

| $\lambda$ | 0 | 0.1 | 0.2 | 0.3 | 0.4 | 0.5 | 0.6 | 0.7 | 0.8 | 0.9 | 1 |
|---|---|---|---|---|---|---|---|---|---|---|---|
| scTM | 0.85 | 0.57 | 0.49 | 0.37 | 0.27 | 0.31 | 0.42 | 0.52 | 0.49 | 0.68 | 0.74 |
| TM-LEFT | 1.0 | 0.78 | 0.66 | 0.61 | 0.51 | 0.35 | 0.36 | 0.38 | 0.40 | 0.42 | 0.43 |
| TM-RIGHT | 0.57 | 0.55 | 0.56 | 0.48 | 0.47 | 0.43 | 0.44 | 0.52 | 0.61 | 0.71 | 1.0 |

