# OpenReview forum: "A Latent Diffusion Model for Protein Structure Generation"
_TMLR — Rejected by TMLR_

### Review · Reviewer_t3NY · 2023-06-03

**Summary Of Contributions:**

Latent Diffusion Model is a proposed method for generating protein structures aiming to reduce the complexity of protein modeling while capturing the distribution of natural protein structures in a condensed latent space. It combines a 3D graph autoencoder and a latent 3D diffusion model to reduce the modeling space of protein structures. The model solves the Bayesian inverse problem of deriving the underlying data distribution by establishing a bijective mapping between given prior distributions and p(z).
To generate a novel protein backbone structure using the Latent Diffusion Model, the process is decomposed into two stages: protein latent representation generation and latent representation decoding. In the first stage, multivariate Gaussian noise is sampled and the learned latent diffusion model is used to generate 3D positions and node embeddings in the latent space. In the second stage, the pre-trained decoder is used to generate backbone structures in the protein space.
Empirically, the Latent Diffusion Model is an effective and efficient approach for generating designable protein backbone structures by combining a 3D graph autoencoder and a latent 3D diffusion model.


**Audience:**

Yes

**Claims And Evidence:**

Yes

**Requested Changes:**

For reproducing the results, uploading the source code and experimental files is important.

**Strengths And Weaknesses:**

# Novelty

The novel aspect of this model is the proposal of a latent diffusion model for generating protein structures. The model reduces the complexity of protein modeling while capturing the distribution of natural protein structures in a condensed latent space, making it more efficient and able to generate novel protein structures effectively. The method combines a 3D graph autoencoder and a latent 3D diffusion model to reduce the modeling space of protein structures, and it ensures rotation and reflection equivariance in the autoencoder design and accurately reconstructs intricate connection information in 3D graphs during decoding. The proposed method can effectively generate novel protein backbone structures with high designability and efficiency, which is important for many synthetic biology applications.

# Advantage

 One advantage is the reduction of the complexity of protein modeling while capturing the distribution of natural protein structures in a condensed latent space. This can make protein modeling more efficient and cost-effective. Another advantage is the high designability and efficiency in generating novel protein backbone structures. Additionally, the proposed method ensures rotation and reflection equivariance in the autoencoder design and accurately reconstructs intricate connection information in 3D graphs during decoding, which improves the accuracy and effectiveness of the model. Finally, the proposed method is more memory-efficient and can demonstrate superiority in sampling speed compared to performing diffusion directly in the protein space.

# Drawback

The limitations of the proposed model include the need to pad input protein backbone structures to a fixed length, the inability to use corresponding generated sequences, and the fact that the 3D protein autoencoder was not trained on all protein structures predicted by AlphaFold.

---

> ### Author Response · Authors · 2023-06-03
> **Response to Reviewer t3NY**
>
> Thank you for your positive feedback and valuable comments! We hope your concerns and questions can be addressed by our responses below.
>
> >**Q1: The limitations of the proposed model include the need to pad input protein backbone structures to a fixed length.**
>
> Our method currently generates proteins with a maximum length of 128, following the same setting as ProtDiff and FoldingDiff. For the convenience of the design and implementation of the 3D protein autoencoder, we pad the length of input protein backbone structures to 128 when training the autoencoder. During generation, the actual generated length is predicted during the decoding process of the protein autoencoder. Specifically, the decoder predicts the end position in the generated protein backbone structure and drops these atoms that appear after the end position. For future work, we will extend the protein autoencoder design to support arbitrary length input and generate arbitrary length protein backbone structure directly without using an end position indicator.
>
> >**Q2: The inability to use corresponding generated sequences.**
>
> Although our 3D protein autoencoder is trained by reconstructing both protein backbone structures and sequences, we do not use the decoded sequences from the autoencoder as generated protein sequences. We clarify the design choice as follows.
>
> Let's use $\boldsymbol{a}$ and $\boldsymbol{x}$ to denote the protein sequences and structures, respectively. Although we could generate protein sequences and structures simultaneously according to the joint probability $p(\boldsymbol{a}, \boldsymbol{x})$, previous studies have shown that it is better to decompose the whole generation process into two stages, such that $p(\boldsymbol{a}, \boldsymbol{x}) = p(\boldsymbol{a}|\boldsymbol{x})p(\boldsymbol{x})$. This means the generative model is only responsible for generating the protein structures, whereas the corresponding protein sequences can be predicted from generated structures as a post hoc process. The reasons are that 1) there already exist good inverse folding models, such as ProteinMPNN, to accurately predict protein sequences from given structures, and 2) two-stage generation can improve the consistency between the generated sequences and structures compared with single-stage generation.
>
> Therefore, we choose the two-stage generation strategy, where we only keep the generated backbone structures from the decoder, and the corresponding sequences are predicted from generated structures by the ProteinMPNN model.
>
> >**Q3: The 3D protein autoencoder was not trained on all protein structures predicted by AlphaFold.**
>
> Our method draws inspiration from stable diffusion that achieved remarkable success in text-to-image generation. A key component of stable diffusion involves pretraining a robust autoencoder model on a vast amount of data (5 billion data). We believe that a similar strategy can be employed for 3D protein generation. Benefiting from the success of Alphafold, deriving accurate protein structure is much easier. AlphaFold Protein Structure Database (AlphaFold DB) has provided over 200 million protein structure predictions. However, due to limited computation resources, we are unable to use all the protein structures from AlphaFold DB for training our 3D protein autoencoder. Consequently, we only incorporate a very small portion of data from AlphaFoldDB, in addition to the commonly used protein data bank. Nevertheless, we have already demonstrated the effectiveness of the proposed method, and we expect performance to be further improved if more data are used, but more computational resources are also needed.
>
> >**Q4: For reproducing the results, uploading the source code and experimental files is important.**
>
> We will release the code after the anonymous review process. In addition, we will provide the pre-trained model parameters for the 3D protein autoencoder and latent diffusion model for reproducing the results in the paper.

---

### Review · Reviewer_rNiM · 2023-06-06

**Summary Of Contributions:**

This paper proposes a method to reduce the protein structure generative space by leveraging the downsampling and upsampling of the amino acids in the protein. Then, with diffusion, the generation could be performed in the reduced space.

**Audience:**

Yes

**Claims And Evidence:**

Yes

**Requested Changes:**

Refer to "Weaknesses & Questions".

**Strengths And Weaknesses:**

### Strengths:
- The proposed method appears to be simple and straightforward.

### Weaknesses & Questions:
- The motivation for the subsampled node seems weak, as its purpose is to improve efficiency. However, many designed proteins are small, typically ranging from 50 to 200 amino acids, and smaller proteins are generally easier to design, synthesize, and analyze experimentally. The proposed model also uses a protein length of $m=128$. With such a small dimension, further compression to something like $m/4$ may not be necessary in practice.

- Why the proposed reduced space method can help to improve the performance? It seems that is not well aligned with the motivation for efficiency.

- The model does not appear to be end-to-end. It first trains the encoder-decoder, then fixes the weight of the encoder-decoder, and finally trains the latent diffusion. Additionally, ProteinMPNN is required for amino acid prediction.

- The random split for train/val/test data seems unreasonable. It would be better to split based on protein similarity. Section A.2 is also confusing; for instance, it mentions "sequence length between 40 and 128 residues" and then "sequence lengths longer than 40."

- The proposed space-reducing method might be more effective as a plugin for existing models, like RFDiffusion. Introducing a new model (encoder-decoder) makes it difficult to determine whether the gain comes from the proposed space-reducing method or the new model.

- In Section 4.3, it is unclear how the 780 backbone structures were generated. Were they generated randomly? How do the baselines generate these backbones, and is the comparison fair? For example, do the models train on the same dataset?

- In Figure 5, the CA-CA distance and bond angle appear unnatural and quite differ from the ground truth values.

- The provided benchmark appears to be inadequate, as it only showcases unconditional single-chain generation. Numerous intriguing tasks in RFDiffusion have not been assessed or explored.

---

> ### Author Response · Authors · 2023-06-20
> **Response to Reviewer rNiM -- Part 1**
>
> Thank you for your time and valuable comments! We hope your concerns and questions can be addressed by our responses below.
>
> >**Q1: The motivation for the subsampled node seems weak, as its purpose is to improve efficiency. However, many designed proteins are small, typically ranging from 50 to 200 amino acids, and smaller proteins are generally easier to design, synthesize, and analyze experimentally. The proposed model also uses a protein length of $m=128$. With such a small dimension, further compression to something like $m/4$ may not be necessary in practice.**
>
> - Thanks for the question. Generating samples in latent space can improve generation efficiency in terms of parallel sampling, as the diffusion model usually needs thousands of steps to generate a sample, and the data must be the same size during reverse steps. Generation in latent space can improves memory efficiency as the latent space is much smaller than the protein space, which enables more efficient parallel sampling under the same computing resources. In real-world applications, it typically requires sampling a large amount of proteins in the screening procedure, so high throughput sampling is always desired. In this sense, parallel sampling in latent space demonstrates significant efficiency improvement even for sampling proteins with a length of 128. For example, we test the generation time of 1000 protein structures in the protein space and latent space using our model, with a single NVIDIA 2080Ti GPU. It takes about 2.9 hours in the protein space, while it only takes about 11 minutes to generate in the latent space and then map to the protein space. So reducing modeling space demonstrates potential usefulness in practice. We also updated Section 4.7 in the revision to clarify the description of parallel sampling efficiency and provide more experiments related to parallel sampling efficiency.
>
> | Generation space | Sampling time (1000 samples) |
> | ----- | ----- |
> | `Protein space` | ~2.9 hrs |
> | `Latent space` | ~11 min |
>
> - We agree that generating longer protein is very important, and we hope to improve this point in future work, but this also requires more computing resources to train the model.
>
>
> >**Q2: Why the proposed reduced space method can help to improve the performance? It seems that is not well aligned with the motivation for efficiency.**
>
> - Actually, besides improving efficiency, another important motivation of reducing modeling space is that it can make the diffusion model easier to learn the desired distribution, as the modeling capacity of diffusion models has a direct relationship with the size of their modeling space. By decreasing the modeling space, we aim to focus the generative model's attention on a more condensed space that relevant to the original protein structure space. This reduction in complexity allows the model to more effectively learn and capture the underlying distribution of protein structures, resulting in improved generation quality.
>
> - Moreover, a smaller modeling space helps address the challenge of high dimensionality and sparsity that is prevalent in protein structure data compared with small molecules. The vast space of possible protein conformations presents a considerable challenge for generative models to learn from limited training data. By narrowing down the modeling space, we provide the generative model with a more manageable and structured latent search space, enabling it to learn the essential features and patterns of protein structures more efficiently.
>
> - We added the clarification to elaborate the motivation of reducing modeling space in the revision at Section 3.1.
>
> >**Q3: The model does not appear to be end-to-end. It first trains the encoder-decoder, then fixes the weight of the encoder-decoder, and finally trains the latent diffusion. Additionally, ProteinMPNN is required for amino acid prediction.**
>
> - Our model is not end-to-end, the major reason is that the distribution is always changing in the latent space during training if the autoencoder and diffusion model are jointly trained, which makes it hard to reflect the real protein structure distribution in the latent space and will break the learning objective of diffusion model. So we need to establish a fixed mapping function from protein space to latent space by training the protein autoencoder as the first stage (similar to pretrained image autoencoder in stable diffusion for text-to-image generation).
>
> - Most previous works, including ProtDiff, FoldingDiff, and RFdiffusion, use ProteinMPNN to predict amino acid sequences from generated protein structures. ProteinMPNN is the state-of-the-art inverse folding model that can predict protein sequences with very high accuracy. In addition, predicting sequences from generated structures can improve the consistency between generated structures and sequences compared with generating structures and sequences together. We added more discussion on this in the revision Section 5.

---

> > ### Comment · Reviewer_rNiM · 2023-06-27
> > **Regards to efficiency (Q1)**
> >
> > It appears that the considerable time cost of length 128 stems from the O(n^2) complexity in the proposed fully-connected graph network architecture, rather than being related to GPU memory limitations. This can be observed from the equation (2.9hrs / 11min = (128/32)^2). When sampling a single protein, the GPU is not operating at full capacity, which means that the time cost may be nearly identical for both lengths 128 and 32. This raises the question: is the fully connected graph truly necessary, given the significant slowdown in the sampling process?

---

> > > ### Author Response · Authors · 2023-06-30
> > > **Response about efficiency**
> > >
> > > Thanks for the pointing it out. Our method can reduce both the memory usage and computational complexity, and thus improve the generation throughput (we revised manuscript to mention complexity). In terms of employing a fully connected graph, we are motivated by the necessity to account for interactions among all atoms. This approach is echoed in numerous protein modeling endeavors, such as RFdiffusion and AlphaFold2, which also take into consideration pairwise amino acid interactions throughout the entire protein. This choice is underpinned by several rationales. Firstly, proteins often engage in long-range interactions that are integral to their structure and function. A fully connected graph facilitates the representation of interactions between all combinations of atoms, regardless of their spatial separation, thereby capturing these long-range interactions. Additionally, proteins are typified by intricate and varied interactions, encompassing electrostatic interactions, hydrogen bonds, Van der Waals forces, and more. The utilization of fully connected graphs offers a more comprehensive representation of the data, enabling graph neural networks to take advantage of the complete connectivity to learn complex patterns and diverse interactions within the protein.

---

> ### Author Response · Authors · 2023-06-20
> **Response to Reviewer rNiM -- Part 2**
>
> >**Q4: The random split for train/val/test data seems unreasonable. It would be better to split based on protein similarity. Section A.2 is also confusing; for instance, it mentions "sequence length between 40 and 128 residues" and then "sequence lengths longer than 40."**
>
> - Following previous works, such as Foldingdiff, we also adopt random split for train/val/test data. For the protein property prediction task, splitting the dataset based on protein similarity might be important, as proteins with high sequence similarity tend to have similar properties, so randomly split can potentially cause "data leakage". However, for the protein generation task, splitting the dataset based on protein similarity is not necessary. The reason is that the protein generation task is to learn the distribution of the protein structure space, and the protein structure space is not related to protein similarity. Therefore, we think the random split is reasonable for the protein generation task.
>
> - Sorry for causing the confusion. In Section A.2, we describe how to curate the dataset from Protein Data Bank (PDE) and AlphaFold Protein Structure Database (AlphaFold DB). There are two parts of data we considered from PDB. The first part is the protein data with **orginal sequence length between 40 and 128 residues** and CA-CA distance less than 5 Å. This part of data is directly added to our dataset. The second part of data we considered are those with **sequence lengths longer than 40** and CA-CA distance less than 5 Å, **but we didn't directly add them in the dataset**. For the second part of the data, we further split them into several fragments at the position where CA-CA distance is larger than 5Å, and then add fragments with length between 50 and 128 into the dataset. We modified the description of section A.2 in the revision to make it more clear.
>
> >**Q5: The proposed space-reducing method might be more effective as a plugin for existing models, like RFDiffusion. Introducing a new model (encoder-decoder) makes it difficult to determine whether the gain comes from the proposed space-reducing method or the new model.**
>
> - RFdiffusion uses pretrained RoseTTAFold as the denoising network and then finetune on the generative tasks. RoseTTAFold needs to be trained on the protein space as it is a protein folding prediction model, so the RFdiffusion cannot be naturally used to generate samples in the latent space.
>
> - As answered in Q2, reducing modeling space can make the diffusion model easier to learn the desired distribution because the generative model's attention can focus on a more condensed space and learn the essential features and patterns of protein structures more efficiently. The comparison with ProtDiff can somewhat validate this. The denoising network we use is similar to ProtDiff, and the big difference is that our generation is performed on a compact latent space. In that sense, our superior performance over ProtDiff demonstrates the effectiveness of reducing the modeling space.
>
> >**Q6: In Section 4.3, it is unclear how the 780 backbone structures were generated. Were they generated randomly? How do the baselines generate these backbones, and is the comparison fair? For example, do the models train on the same dataset?**
>
> - We generate these 780 backbone structures randomly. We split the generated proteins into two length categories, following ProtDiff and FoldingDiff, and we compare the success rate of design (scTM > 0.5) within each length category. The success rates of baselines are obtained from the FoldingDiff paper. Our model didn't train on the same dataset as theirs. For the baseline models, their dataset only contains about 4k (ProtDiff) or 24k (FoldingDiff) training data. However, since we want to build a condensed and well-structured latent space, a large amount of protein data is desired to train the protein autoencoder (just like in stable diffusion, 5 billion data are used to pretrain a robust image autoencoder). In this case, we need to augment the training data by including more protein structures from the AlphaFold Protein Structure Database (AlphaFold DB).

---

> > ### Comment · Reviewer_rNiM · 2023-06-27
> > **Regards to fair comparison (Q6)**
> >
> > In Table 2, we observe that the performance follows the trend "latentdiff > foldingdiff > protdiff," which is consistent with the training data size. This suggests that the performance gain may be attributed to the larger dataset. To confirm this hypothesis, I recommend conducting an ablation study with equal data sizes for a more accurate comparison.

---

> > > ### Author Response · Authors · 2023-06-30
> > > **Response about fair comparison**
> > >
> > > Fundamentally, our methodology necessitates that the latent space is well-structured and equivalent with the protein space. To realize this goal to the maximum extent, it is imperative that we utilize the maximum possible volume of data to train our model. Consequently, data reduction is not a viable approach within the confines of our methodology. But in response to the reviewer's query, we are utilizing our dataset to train the ProtDiff. The time is limit, but we are trying our best to provide this result before the rebuttal deadline.

---

> > > > ### Author Response · Authors · 2023-07-06
> > > > **Further response on fair comparison**
> > > >
> > > > Thank you for your patience. During the training of ProtDiff with our dataset, we noticed that the loss was decreasing very slowly. Consequently, we experimented with a smaller dataset, which, at approximately 17k, is still much larger than what ProtDiff utilized. With this adjustment, we observed a faster decrease in loss. However, as of now, the performance of ProtDiff has not converged to the same level as the original one. We hypothesize that ProtDiff may have been prone to overfitting on their original dataset given that their dataset was relatively small with only 4k entries. Thus, introducing more data should have mitigated the overfitting and model now might require more capacity and a longer training duration to adapt effectively. Moreover, the additional data might have increased the diversity within the dataset, potentially posing challenges for the ProtDiff in learning the distribution effectively. As a compromise, we have revised the manuscipt to underscore the necessity of employing a larger dataset to train our model and to explicitly alert readers to the disparity in dataset sizes relative to other methods.

---

> > > > > ### Comment · Reviewer_rNiM · 2023-07-08
> > > > > **Thank you for the response.**
> > > > >
> > > > > Thank you for taking the time to run ProtDiff on larger datasets; your observations are indeed quite intriguing. However, I would suggest running LatentDiff on smaller datasets, as you are already well-versed in tuning LatentDiff.
> > > > >
> > > > > Due to the time constraints, I understand that conducting an ablation study during this discussion period may not be feasible. Nevertheless, I strongly recommend incorporating this study in your next revision.

---

> > > > > > ### Author Response · Authors · 2023-07-11
> > > > > > **Thank you for your understanding and suggestion.**
> > > > > >
> > > > > > Thank you for your suggestion, and we appreciate your understanding of the time constraints we are facing during the discussion period. We are currently planning to conduct an ablation study concerning the impact of data size. This will take some time, but we intent to include them along with some discussion in the final camera-ready version of our work.
> > > > > >
> > > > > > Specifically, we are still training ProtDiff with larger datasets to investigate whether there will be an improvement in performance and to compare it with LatentDiff. Furthermore, in line with the reviewer's suggestion, we plan to train LatentDiff using a smaller PDB dataset (~4k) that was previously used by ProtDiff. With respect to the use of the PDB data, we are considering two options: Firstly, we can train both the 3D protein autoencoder and the latent diffusion model using the small PDB dataset. Alternatively, we can pretrain the 3D protein autoencoder with our own large dataset to optimally ensure a well-structured latent space and good reconstruction performance. Following this, we could train the latent diffusion model with the small PDB data. By exploring these options, we aim to gain a deeper understanding of how data size influences each component of LatentDiff.
> > > > > >
> > > > > > Thank you again for your insightful feedback.

---

> ### Author Response · Authors · 2023-06-20
> **Response to Reviewer rNiM -- Part 3**
>
> >**Q7: In Figure 5, the CA-CA distance and bond angle appear unnatural and quite differ from the ground truth values**
>
> - Figure 5 shows the distribution of CA-CA distance and bond angle calculated over all generated protein structures and all test set protein structures. We acknowledge that due to the nature of the generative model, it is unlikely for these values to be identical to the ground truth. As a generative model, the objective is to capture the underlying patterns and statistical properties of protein structures rather than replicating exact ground truth values. Moreover, maximum likelihood training of generative models optimizes the model towards mean-seeking and mode-covering direction, so the learned distribution tends to become smoother than the ground truth distribution. In that sense, the CA-CA distance and bond angle in Figure 5 already exhibit a notable resemblance to the natural protein distribution, even though not perfect.
>
> - As a future work, we will improve the generative model to make the model and data distribution fit even better.
>
> >**Q8: The provided benchmark appears to be inadequate, as it only showcases unconditional single-chain generation. Numerous intriguing tasks in RFDiffusion have not been assessed or explored.**
>
> - RFdiffusion is a very promising method in protein generation and outperforms all previous methods by a large margin. In the revision, we also added a discussion with RFdiffusion in Section 2.3. RFdiffusion leverages the powerful protein structure prediction model, RoseTTAFold, to achieve remarkable results on many generation tasks. RFdiffusion pretrains RoseTTAFold on protein structure prediction task for 4 weeks with 64 V100 GPUs, and then uses pretrained RoseTTAFold as the denoising network to finetune on the generative tasks. And RFdiffusion only demonstrates the effectiveness of generating proteins when using pretrained weights.
>
> - In our work, we only focus on unconditional generation, the most fundamental task of protein generation, and we want to explore a new methodology that possesses great potential to better solve this problem. These conditional generation tasks are very exciting to be explored, but due to the scope of this paper, we plan to leave them for future research.

---

### Review · Reviewer_y3vc · 2023-06-08

**Summary Of Contributions:**

Diffusion models are quite popular for generating images, but generation is quite computationally demanding. Recent work has sped up generation by using the diffusion model to sample in the latent space from a pretrained image autoencoder (e.g., stable diffusion). Diffusion models have recently shown promise for sampling 3D protein structures, and this paper explores the potential for generation costs to be reduced using a pretrained 3D structure autoencoder.

Much of the technical content of the paper is devoted to ensuring that the autoencoder and the diffusion model are equivariant to 3D rotations, flips, and translations.

Experimental results demonstrate that samples from the model have similar quality to recent work, but are substantially faster.


**Audience:**

Yes

**Claims And Evidence:**

Yes

**Requested Changes:**

Please addresses all of my comments in 'Weaknesses' above except 'sequence length'.


**Strengths And Weaknesses:**

# Strengths
This is a useful contribution to the research dialog on diffusion models for proteins. Given the success of latent diffusion for images, many protein researchers will be curious to see how it generalizes to proteins.

Being able to generate diverse novel proteins has broad interest in drug design, biocatalysis, etc.

# Weaknesses

## Evaluation
As the authors explain, the gold standard evaluation setup would require synthesizing new proteins and measuring their structures. This would be extremely expensive, so this work (and prior work) depends on in-silico surrogate metrics. With this in mind, I found that the description of this setup lacking in key details:
Can you explain what the train-test split is? How would a model perform that just memorized a single training example? As far as I understand, it would do well under the criteria in Table 2. It's possible that your model has better percentage of samples with scTM > 0.5, but that these samples are substantially less diverse than prior work.
You write that the 'maximum TM-score among these eight scores is referred to as the self-consistency TM-score (scTM).' Can you please explain why taking this maximum is appropriate? In practice, if one was designing a protein, how would they choose among samples?
Sec 4.3: Can you explain why you chose to use omegafold as your structure prediction model? Is this what the other systems in Table 2 used? If not, could this be an unnecessary confounder when comparing the systems?

The visual analysis of samples was underwhelming. The paper writes: "we also visualize some exemplar backbones and OmegaFold-predicted backbone structures using PyMOL (DeLano, 2002) in Figure 4, which clearly shows that our LatentDiff is capable of generating designable proteins". These are just a few examples for which scTM is low, so it doesn't provide new information over the previous tables. I would have appreciated, for example, a demonstration that the samples capture a diverse set of folds, secondary structure elements.

## Exposition
I found the description of the encoder/decoder architectures quite confusing. It wasn't clear to me why the language of graph neural networks is necessary if the graph is fully connected and some of the update rules are convolutional. For example, why describe 'edge building' when the graph is fully connected? Could it be expressed in terms of CNN and MLP layers? Is it some sort of transformer?

## Related work
I was very surprised that you didn't discuss the RFDiffusion paper: Watson et al. "Broadly applicable and accurate protein design by integrating structure prediction networks and diffusion generative models." Similarly, I was surprised by how little Ingraham et al. was discussed. Can you please discuss the relationship between your work and theirs?

## Speedup Results
I'm very confused as to how a speedup is obtained by your system. The encoder-decoder model was tuned to provide a downsampling factor of 4. At first, it seems that the diffusion model would be generating data that is 4x lower-dimensional. However, the diffusion model also needs to generate not just X, but also H, so there are 4x fewer nodes in the graph, but each node has more dimensions. Can you please explain how such a big speedup was obtained in section 4.5? Are you comparing results from two different software packages, or did you implement ProtDiff yourself?

## Sequence Length
As with other recent work on diffusion for proteins, the authors can't generate proteins longer than ~128 amino acids, which is considerably shorter than most natural proteins. I expected that the latent-space sampling would enable generating longer sequences, but was disappointed to see that this was explored.

---

> ### Author Response · Authors · 2023-06-20
> **Response to Reviewer y3vc-- Part 1**
>
> Thank you for your time and valuable comments! We hope your concerns and questions can be addressed by our responses below.
>
> >**Q1: As the authors explain, the gold standard evaluation setup would require synthesizing new proteins and measuring their structures. This would be extremely expensive, so this work (and prior work) depends on in-silico surrogate metrics. With this in mind, I found that the description of this setup lacking in key details: Can you explain what the train-test split is? How would a model perform that just memorized a single training example? As far as I understand, it would do well under the criteria in Table 2. It's possible that your model has better percentage of samples with scTM > 0.5, but that these samples are substantially less diverse than prior work. You write that the 'maximum TM-score among these eight scores is referred to as the self-consistency TM-score (scTM).' Can you please explain why taking this maximum is appropriate? In practice, if one was designing a protein, how would they choose among samples? Sec 4.3: Can you explain why you chose to use omegafold as your structure prediction model? Is this what the other systems in Table 2 used? If not, could this be an unnecessary confounder when comparing the systems?**
>
> 1. *Can you explain what the train-test split is?*
> - We randomly split the dataset into training, validation, and test set. The ratio of training, validation, and test set is 0.8, 0.1, and 0.1, respectively. But for the in-silico evaluation, we don't need to use test set as the scTM score is calculated solely based on the generated proteins. The test set is only used for the evaluation of some statistical distributions calculated from the generated proteins.
>
> 2. *It's possible that your model has better percentage of samples with scTM > 0.5, but that these samples are substantially less diverse than prior work.*
> - In Appendix A.5, we evaluate the diversity of generated proteins with scTM > 0.5 (designable), and we moved this to the main body, Section 4.6, in the revision. Specifically, for each designable protein, we calculate the TM score with all other designable proteins and choose the maximum TM score to measure the similarity with other generated proteins. And then, we calculate the average of maximum TM scores overall designable proteins to measure the diversity of generated proteins. The results are shown in the table below. We can see that our method achieves the diversity between ProtDiff and FoldingDiff, which demonstrate that our method can generate diverse designable proteins instead of memorizing a single training example (in that case, the diversity score will be 1).
>
> | Method | Diversity $\downarrow$ |
> | ----- | ----- |
> | `ProtDiff` | 0.836 $\pm$ 0.1648 |
> | `FoldingDiff` | 0.585 $\pm$ 0.1276 |
> | `LatentDiff` | 0.615 $\pm$ 0.0849 |
>
>
> 3. *You write that the 'maximum TM-score among these eight scores is referred to as the self-consistency TM-score (scTM).' Can you please explain why taking this maximum is appropriate? In practice, if one was designing a protein, how would they choose among samples?*
>
> - We use ProteinMPNN to predict amino acid sequences from generated protein structures. We follow ProtDiff and FoldingDiff to sample 8 amino acid sequences for each generated protein and choose the one that is most likely to fold into the generated backbone structure (i.e. the predicted structure for that sequence best match the generated structure than other sequences in terms of TM-score). We think choosing the maximum TM score is reasonable since we want to find whether the generated structures can be designed (only amino acid sequences can be made in the wet lab), so we need to choose the most likely sequence. In practice, if one was designing a protein, one should also choose the amino acid sequence with the highest TM score as this give the best chance to successfully make the generated backbone structure in the wet lab.
>
> 4. *Sec 4.3: Can you explain why you chose to use omegafold as your structure prediction model? Is this what the other systems in Table 2 used? If not, could this be an unnecessary confounder when comparing the systems?*
> - We follow FoldingDiff to use OmegaFold as the structure prediction model when calculating the scTM score. The reason is that AlphaFold2 inference is too slow due to the MSA search, whereas OmegaFold is orders of magnitude faster than AlphaFold2. In addition, OmegaFold can achieve a very similar performance as AlphaFold2 based on the results in their paper. So we think OmegaFold is feasible to serve as a structure prediction model in calculating scTM score.

---

> ### Author Response · Authors · 2023-06-20
> **Response to Reviewer y3vc-- Part 2**
>
> >**Q2: The visual analysis of samples was underwhelming. The paper writes: "we also visualize some exemplar backbones and OmegaFold-predicted backbone structures using PyMOL (DeLano, 2002) in Figure 4, which clearly shows that our LatentDiff is capable of generating designable proteins". These are just a few examples for which scTM is low, so it doesn't provide new information over the previous tables. I would have appreciated, for example, a demonstration that the samples capture a diverse set of folds, secondary structure elements.**
>
> - Our method can generate proteins with multiple secondary structure elements, just as what appears in natural proteins. We use P-SEA to count the number of two kinds of important secondary structures ($\alpha$-helix and $\beta$-sheets) for generated proteins with scTM > 0.5. Specifically, we calculate the percentage of generated proteins that contain only $\alpha$-helix, only $\beta$-sheets, and both $\alpha$-helix and $\beta$-sheets, as shown in the table below. More than half of the generated proteins include $\alpha$-helix, and a large portion of generated proteins contain $\beta$-sheet. We move this experiment from Appendix to Section 4.5 in the revision.
>
> | Method | $\alpha$-helix only | $\beta$-sheets only | $\alpha$-helix +  $\beta$-sheets|
> | ----- | ----- | ----- | ----- |
> | `LatentDiff` | 58.7% | 13.3% | 14.9% |
>
> - However, there aren't too many very different folds in the generated proteins, part of the reason could be that the length of generated proteins is limited. We deleted that sentence in the revision if it might sound misleading or exaggerated to the reviewer. And we hope to improve this point about diverse folds in our future research.
>
> >**Q3: I found the description of the encoder/decoder architectures quite confusing. It wasn't clear to me why the language of graph neural networks is necessary if the graph is fully connected and some of the update rules are convolutional. For example, why describe 'edge building' when the graph is fully connected? Could it be expressed in terms of CNN and MLP layers? Is it some sort of transformer?**
>
> - Sorry for causing the confusion. We use the language of graph neural network because (1) In our method, proteins are represented as graphs with 3D geometries (each node has a 3D position), and (2) We do use a graph neural network (EGNN) to perform message passing on the graph data to update node position and node feature.
> - In addition, we need to make some clarification: (1) Within each downsampling or upsampling stage, the **edge building** and **graph expansion** steps are performed sequentially to construct a graph as input to the graph neural network. The message passing happens after these two steps. Moreover, for the downsampling, we didn't construct fully connected graph in the **edge building** step, where padding nodes are not included in the graph after this step (but will connect to the downsampling nodes in the following **graph expansion** step). (2) In addition, even if the graph is fully connected after **edge building** step, in the **graph expansion** step, the initialized downsampled nodes will be connected to the graph, and the new graph is not fully connected anymore before we conduct message passing to update the graph. So this can not be expressed as CNN or pure MLP layers (although message passing involves some MLP layers).
>
> - We revised the description of the encoder/decoder architectures in the revision to make it more clear.

---

> ### Author Response · Authors · 2023-06-20
> **Response to Reviewer y3vc-- Part 3**
>
> >**Q4: I was very surprised that you didn't discuss the RFDiffusion paper: Watson et al. "Broadly applicable and accurate protein design by integrating structure prediction networks and diffusion generative models." Similarly, I was surprised by how little Ingraham et al. was discussed. Can you please discuss the relationship between your work and theirs?**
>
> - Thanks for pointing out RFdiffusion and Chroma. In parallel with the development of our LatentDiff, we noticed that these two new models of protein production were also developed. RFdiffusion and Chroma outperform all the previous work by a large margin, enabling generating long proteins with very high quality, and they both also explored lots of conditional generation tasks.
>
> - RFdiffusion takes advantage of the powerful protein structure prediction model, RoseTTAFold, to achieve remarkable results on many generation tasks. RFdiffusion pretrains RoseTTAFold on protein structure prediction task for 4 weeks with 64 V100 GPUs. It then uses pretrained RoseTTAFold as the denoising network to finetune on the generative tasks. And RFdiffusion only demonstrates the effectiveness of generating proteins only when using pretrained weights. Chroma uses a correlated diffusion process to transform protein structures into random collapsed polymers and encode the chain and radius of gyration constraints by a designed covariance model.
>
> - Essentially, both Chroma and our LatentDiff aim to efficiently model the target distribution. As said in the Chroma paper, the stable diffusion model can be viewed as learning a transformed coordinate system in which the diffusion model can more efficiently model the target distribution, which shares the same motivation with our LatentDiff model. Whereas Chroma uses correlated diffusion to add the constraint to the noise process so that the diffusion model does not need to learn basic structure constraints, and thus the network can model distribution more efficiently.
>
> - We added a paragraph to discuss these two works in Section 2.3 in the revision.
>
> >**Q5: I'm very confused as to how a speedup is obtained by your system. The encoder-decoder model was tuned to provide a downsampling factor of 4. At first, it seems that the diffusion model would be generating data that is 4x lower-dimensional. However, the diffusion model also needs to generate not just X, but also H, so there are 4x fewer nodes in the graph, but each node has more dimensions. Can you please explain how such a big speedup was obtained in section 4.5? Are you comparing results from two different software packages, or did you implement ProtDiff yourself?**
>
> - First, we need to clarify that even if a protein is generated in the protein space, such as ProtDiff, each node also has feature vector $\boldsymbol{h}$ besides position $\boldsymbol{x}$. The feature vector is used in the message passing to update the node position, and itself will also be updated.
>
> - We updated Section 4.5 (now Section 4.7) in the revision to make the description of speed comparison more clear (Originally, in Section 4.5, we didn't compare with ProtDiff, we only compared with our own method on generating in protein space without using downsampling. But in the revision, we also compare with ProtDiff, and we use code provided by ProtDiff). We compare the generation efficiency in terms of parallel sampling. Generation in latent space can improve memory efficiency as the latent space is much smaller than the protein space. So for the same amount of GPU memory, we can sample more proteins in latent space than in protein space. The reason we compare efficiency in terms of parallel sampling is that, in practice, it requires sampling a large amount of proteins in the screening procedure, so high throughput sampling is desired. In this sense, sampling in latent space demonstrates significant efficiency improvement. For example, for our diffusion model, the generation time of 1000 protein structures in the protein space is about 2.9 hours, while it only takes about 11 minutes to generate in the latent space and then map to the protein space.

---

> ### Author Response · Authors · 2023-06-20
> **Response to Reviewer y3vc-- Part 4**
>
> >**Q6: As with other recent work on diffusion for proteins, the authors can't generate proteins longer than ~128 amino acids, which is considerably shorter than most natural proteins. I expected that the latent-space sampling would enable generating longer sequences, but was disappointed to see that this was explored.**
>
> - Due to the limited computing resources, we limit the protein length to 128, making it the same as ProtDiff and FoldingDiff. It was demonstrated in stable diffusion that a powerful and robust pretrained autoencoder is a key part towards the success of the latent diffusion model. For example, stable diffusion uses 5 billion images to pretrain a robust image autoencoder. Thus, we believe our LatentDiff has the potential to generate longer proteins, but this requires more training data and computing resources to train a powerful and robust protein autoencoder that can establish a well-structured latent space to map protein structures to latent representations.

---

### Review · Reviewer_NsUN · 2023-06-10

**Summary Of Contributions:**

The authors present a generative approach for protein backbones. The idea is to operate in a similar way to the stable diffusion approach for image generation, i.e. apply diffusion in the latent space. The authors propose to do the same for 3D graphs. They identify the following challenges: 1) E(3) equivariance for the autoencoder, 2) accuracy of decoding for 3D graphs, 3) diffusion process should be specialised for 3D.
The contributions are: 1) a novel E(3) graph autoencoder,  2) a novel E(3) diffusion process, 3) an overall increase in sampling efficiency due to operating in a reduced latent space.

**Audience:**

Yes

**Broader Impact Concerns:**

The capacity to generate biological agents and filter them with computational approaches to select potentially dangerous agents should be considered.

**Claims And Evidence:**

No

**Requested Changes:**

The paper proposes an efficient approach for the generation of protein backbones. The efficiency aspect is underdeveloped in the paper and the relative experimental results receive only 10 line in Section 4.5 and one numerical comparative result.
Important questions are, how does the computational efficiency of the proposed approach scale with (and compare against other systems):
1. the length of the protein (the parameter m)
2. the number of latent nodes?
3. the number of layers? the number of steps in the diffusion process?
4. the overall number of parameters of the whole architecture?

To reiterate a previous point: since the latent representation consists of a small number of nodes in 3D space, it would be of interest to visualise these latent embeddings and identify trends, clustering properties, relationships to functional annotations of the associated proteins, etc.

**Strengths And Weaknesses:**

The idea of working in a latent space that is subject to constraints such as E(3) is interesting. The authors should however show that this brings advantages with respect to other approaches, for example working with nodes with attributes of a different number of dimensions and not forced to be rotation and reflection invariant.
Given that the latent representation consists of a small number of nodes in 3D space, it would be of interest to visualise these latent embeddings and identify trends, clustering properties, relationships to functional annotations of the associated proteins, etc. This seems to be an important opportunity that is completely missed.

The authors offer a 3D equivariant graph autoencoder, but several other approaches exist in literature and these are not acknowledged nor compared to; why is the proposed approach better than other? e.g.
1. Batzner, Simon, Albert Musaelian, Lixin Sun, Mario Geiger, Jonathan P. Mailoa, Mordechai Kornbluth, Nicola Molinari, Tess E. Smidt, and Boris Kozinsky. "E (3)-equivariant graph neural networks for data-efficient and accurate interatomic potentials." Nature communications 13, no. 1 (2022): 2453.
2. Zhang, Yang, Wenbing Huang, Zhewei Wei, Ye Yuan, and Zhaohan Ding. "EquiPocket: an E (3)-Equivariant Geometric Graph Neural Network for Ligand Binding Site Prediction." arXiv preprint arXiv:2302.12177 (2023).

One important aspect seems to be the joint generation of 3D coordinates and attributes (i.e. the amino acid type) but then the authors simply state that:<<Note that we do not use reconstructed amino acid types for the corresponding node. Instead, we use the inverse folding model ProteinMPNN (Dauparas et al., 2022) to predict protein amino acid sequences from generated backbone structures.>> The authors should offer some insights on why the predicted ACs are not of acceptable quality. In the Discussion's future directions the authors state that this is one aspect to improve but do not offer any analysis or suggestion on how to improve this shortcoming.

In literature we see now conditional generative methods that can bias the generation process according to desired prompts/constraints, but the authors propose an unconditional model to simply sample protein structures at random; the authors should justify the use case for their approach.

To improve the overall quality of the paper the authors should consider a clearer introduction of the main ideas, i.e. why using 3D latent nodes is a good idea; the paper does not read smoothly and clearly, its clarity could be significantly improved.

---

> ### Author Response · Authors · 2023-06-20
> **Response to Reviewer NsUN -- Part 1**
>
> Thank you for your time and valuable comments! We hope your concerns and questions can be addressed by our responses below.
>
> >**Q1: The idea of working in a latent space that is subject to constraints such as E(3) is interesting. The authors should however show that this brings advantages with respect to other approaches, for example working with nodes with attributes of a different number of dimensions and not forced to be rotation and reflection invariant.**
>
> - Thanks for the question. We would like to clarify that making the network to be equivariant is not an ad hoc design choice, and it is actually derived from the distribution invariance requirement. Specifically, when a protein structure $\boldsymbol{x}$ is under a rigid body transformation $R$, since the protein structure itself still remains unchanged, the probability $p(R\boldsymbol{x}) = p(\boldsymbol{x})$ must hold. In order to achieve this distribution invariance, the generation process should be equivariant to the transformation $R$, as proved in [1]. In our method, we decompose the generation of protein structures into two stages: (1) use a denoising network to generate latent protein representations and (2) decode latent representations into protein structures. So both of these two stages must be equivariant to the transformation $R$ in order to achieve distribution invariance.
>
> - In addition, we tried adding the KL-divergence penalty on the equivariant feature (3D position) to make it towards invariant, then the reconstruction performance became a lot worse (RMSD 13.73 vs. RMSD 1.41 for equivariant).
>
> [1] Hoogeboom, Emiel, et al. "Equivariant diffusion for molecule generation in 3d." International Conference on Machine Learning. PMLR, 2022.
>
> >**Q2: Given that the latent representation consists of a small number of nodes in 3D space, it would be of interest to visualise these latent embeddings and identify trends, clustering properties, relationships to functional annotations of the associated proteins, etc. This seems to be an important opportunity that is completely missed.**
>
> - Thanks for the question. We provided some discussion about latent space in the Appendix A.4 in the revision. For your convenience, we also include the discussion to this reply:
>
>   - Usually, it is natural to visualize the latent space and perform latent code interpolation to test if the latent space is well-structured. However, a protein in our latent space is not represented by a single latent feature vector, but rather, it is a set of nodes associated with 3D coordinates and node features. As such, it is difficult to use dimension reduction techniques like t-SNE to visualize the latent space. In addition, we did not add a KL-divergence loss on coordinates since it would break equivariance. Even for invariant node features, we only add a minimal KL-divergence penalty to control the variance of the latent space, as we aim to maintain high reconstruction accuracy for the autoencoder. Therefore, in our case, the latent space does not necessarily need to be perfectly well-structured, and arbitrary interpolation may not guarantee valid protein structures upon decoding.
>
>   - We pick two generated proteins with scTM>0.5 (designable), and their corresponding latent space data are $(X^s_{emb}, H^s_{emb})$ and $(X^t_{emb}, H^t_{emb})$. Then we interpolate these two latent space data as $(X^{interp}{emb}, H^{interp}{emb})=(X^s_{emb} *(1-\lambda) + X^t_{emb} * \lambda, H^s_{emb} *(1-\lambda) + H^t_{emb} * \lambda)$. We choose different values of $\lambda$ and decode the interpolated latent space data into proteins and calculate the scTM score, as shown below. TM-left means the TM score with the start protein, and TM-right means the TM score with the end protein. We can see that if $\lambda$ is close to 0 or 1, generated proteins are still designable. However, if $\lambda$ is near 0.5, generated proteins are not valid, just as we analyzed above.
>
> | $\lambda$ | 0 | 0.1 | 0.2 | 0.3 | 0.4 | 0.5 | 0.6 | 0.7 | 0.8 | 0.9 | 1 |
> | ----- | ----- | ----- | ----- | ----- | ----- | ----- | ----- | ----- | ----- | ----- | ----- |
> | `scTM` | 0.85 | 0.57 | 0.49 | 0.37 | 0.27 | 0.31 | 0.42 | 0.52 | 0.49 | 0.68 | 0.74 |
> | `TM-left` | 1.0 | 0.78 | 0.66 | 0.61 | 0.51 | 0.35 | 0.36 | 0.38 | 0.40 | 0.42 | 0.43 |
> | `TM-right` | 0.57 | 0.55 | 0.56 | 0.48 | 0.47 | 0.43 | 0.44 | 0.52 | 0.61 | 0.71 | 1.0 |
>
> - Exploring the relationship between latent space and functional annotations seems to be interesting, and we will leave it for future research.

---

> ### Author Response · Authors · 2023-06-20
> **Response to Reviewer NsUN -- Part 2**
>
> >**Q3: The authors offer a 3D equivariant graph autoencoder, but several other approaches exist in literature and these are not acknowledged nor compared to; why is the proposed approach better than other? e.g. Batzner, Simon, Albert Musaelian, Lixin Sun, Mario Geiger, Jonathan P. Mailoa, Mordechai Kornbluth, Nicola Molinari, Tess E. Smidt, and Boris Kozinsky. "E(3)-equivariant graph neural networks for data-efficient and accurate interatomic potentials." Nature communications 13, no. 1 (2022): 2453. Zhang, Yang, Wenbing Huang, Zhewei Wei, Ye Yuan, and Zhaohan Ding. "EquiPocket: an E(3)-Equivariant Geometric Graph Neural Network for Ligand Binding Site Prediction." arXiv preprint arXiv:2302.12177 (2023).**
>
> - Thanks for pointing out these two papers. In terms of tasks, these two papers actually didn't consider the protein generation problem. In addition, these two works do not consider modeling space reduction, whereas we need to design protein downsampling and upsampling method to reduce modeling space in our protein autoencoder. Thus, they are not directly comparable with our work. Nevertheless, we think these two papers are very interesting and related to equivariant networks. **So we cited these two papers in the revision and discuss some relationships with these two papers here**:
>
>   - Batzner, et al [1] proposed NequIP to learn interatomic potentials from ab-initio calculations for molecular dynamics simulations. EquiPocket [2] used an equivariant graph neural network to predict ligand binding sites. NequIP uses spherical harmonics to ensure equivariance and involves higher-order geometric tensor, whereas we adopt EGNN to perform equivariant message passing that only support type 1 tensor but is more computationally efficient. EquiPocket also uses EGNN to conduct message passing, but it is performed over the surface graph extracted from protein that is tailored to the specific binding site prediction problem they are trying to solve.
>
> [1] Batzner, Simon, Albert Musaelian, Lixin Sun, Mario Geiger, Jonathan P. Mailoa, Mordechai Kornbluth, Nicola Molinari, Tess E. Smidt, and Boris Kozinsky. "E(3)-equivariant graph neural networks for data-efficient and accurate interatomic potentials." Nature communications 13, no. 1 (2022): 2453
>
> [2] Zhang, Yang, Wenbing Huang, Zhewei Wei, Ye Yuan, and Zhaohan Ding. "EquiPocket: an E(3)-Equivariant Geometric Graph Neural Network for Ligand Binding Site Prediction." arXiv preprint arXiv:2302.12177 (2023).

---

> ### Author Response · Authors · 2023-06-20
> **Response to Reviewer NsUN -- Part 3**
>
> >**Q4: One important aspect seems to be the joint generation of 3D coordinates and attributes (i.e. the amino acid type) but then the authors simply state that:"Note that we do not use reconstructed amino acid types for the corresponding node. Instead, we use the inverse folding model ProteinMPNN (Dauparas et al., 2022) to predict protein amino acid sequences from generated backbone structures." The authors should offer some insights on why the predicted ACs are not of acceptable quality. In the Discussion's future directions the authors state that this is one aspect to improve but do not offer any analysis or suggestion on how to improve this shortcoming.**
>
> - Thanks for the question. We elaborate our analysis below and hope this can address your concerns.
>
> - Let's use $\boldsymbol{a}$ and $\boldsymbol{x}$ to denote the protein sequences and structures, respectively. Instead of directly sampling from the joint probability $p(\boldsymbol{a}, \boldsymbol{x})$, previous studies have shown that it is better to decompose the whole generation process into two stages, such that $p(\boldsymbol{a}, \boldsymbol{x}) = p(\boldsymbol{a}|\boldsymbol{x})p(\boldsymbol{x})$. This means the generative model is only responsible for generating the protein structures, whereas the corresponding protein sequences can be predicted from generated structures as a post hoc process. We think the reason of single stage generation not working is that directly modeling the joint distribution of structures and sequences is very hard since the model needs to learn the complex correlation between protein structures and amino acid sequences. This somewhat requires the generative model to implicitly learn the inverse folding or folding process, which by themselves are complex tasks that need powerful inverse folding or protein folding prediction models to solve. In addition, people actually care more about protein structures that define proteins' function, whereas amino acid sequences are just used to make proteins in the wet lab. So if we already have a good inverse folding model that can obtain amino acid sequence, it is better to use it to predict amino acid sequence from generated protein structures. And this will reduce the burden of the generative model to learn the complex correlation between protein structures and amino acid sequences.
>
> - We added the above discussion in the revision Section 5 about why jointly generating 3D structure and sequence does not work well and added some suggestions about how to improve this. We also list the potential solution here:
>
>   - Potential solution 1: An iterative refinement approach involves alternating between sequence and structure generation steps. Initially, a rough structure can be generated from the given sequence, and then the sequence can be optimized to better match the generated structure. This process can be repeated several times, gradually improving the consistency between the generated sequence and structure.
>
>   - Potential solution 2: Incorporate physical constraints during the generative modeling process. By integrating physical principles such as energy functions and geometric constraints into the generative model, it becomes possible to guide the generation process to produce sequences that are more likely to fold into the generated structures.
>
>   - Potential solution 3: With a huge amount of training data to pretrain the protein autoencoder and latent diffusion model, the correlation between structure and sequence might be well captured, and thus, the modeling difficulty of single stage generation might be naturally addressed.

---

> ### Author Response · Authors · 2023-06-20
> **Response to Reviewer NsUN -- Part 4**
>
> >**Q5: In literature we see now conditional generative methods that can bias the generation process according to desired prompts/constraints, but the authors propose an unconditional model to simply sample protein structures at random; the authors should justify the use case for their approach.**
>
> - Thanks for this question, and we provide the clarification on this as below.
>   - While we acknowledge the importance and usefulness of conditional generation, we would like to emphasize the significance of unconditional protein generation. Unconditional generation allows us to explore the diverse space of protein structures, uncovering novel folding patterns, rare motifs, and alternative conformations. It could provide insights into the inherent variability of proteins and serves as a foundation for understanding protein folding principles. Additionally, unconditional generation enables the discovery of functional and stable protein structures, with potential applications in enzyme design, drug discovery, and synthetic biology. The generated structures can also enhance the training data for subsequent conditional models, improving their performance and diversity. Overall, unconditional generation complements conditional generation by providing novel insights, diverse structures, and foundational knowledge.
>
>   - Conditional generation is built on top of unconditional generation capability and, indeed is more practically useful and very exciting to be explored, but due to the scope of this paper, we plan to leave them for future research.

---

> ### Author Response · Authors · 2023-06-20
> **Response to Reviewer NsUN -- Part 5**
>
> >**Q6: To improve the overall quality of the paper the authors should consider a clearer introduction of the main ideas, i.e. why using 3D latent nodes is a good idea; the paper does not read smoothly and clearly, its clarity could be significantly improved.**
>
> - Thanks for the suggestion. In the revision, we added the motivation of reducing modeling space using a 3D protein autoencoder in Section 3.1. Specifically, we elaborate on why reducing modeling space can help improve the generation quality and the efficiency of generation. Also, we summarize the revision in the general response. Hope the manuscript can be more clear to read after revision.
>
> >**Q7: The paper proposes an efficient approach for the generation of protein backbones. The efficiency aspect is underdeveloped in the paper and the relative experimental results receive only 10 line in Section 4.5 and one numerical comparative result. Important questions are, how does the computational efficiency of the proposed approach scale with (and compare against other systems): 1. the length of the protein (the parameter m) 2. the number of latent nodes? 3. the number of layers? the number of steps in the diffusion process? 4. the overall number of parameters of the whole architecture?**
>
> - Thanks for the comments. We added more experiments to test the efficiency of our method in terms of parallel sampling, and we updated the results and writing to the revision Section 4.7. We also respond here:
>
>   - Generation in latent space can improve memory efficiency as the latent space is much smaller than the protein space. So for the same amount of GPU memory, we can sample more proteins in latent space than in protein space. The reason we compare efficiency in terms of parallel sampling is that, in practice, it requires sampling a large amount of proteins in the screening procedure, so high throughput sampling is desired. In this sense, sampling in latent space demonstrates significant efficiency improvement.
>
>   - For the experiments, we compare sampling 1000 proteins with different methods on a single NVIDIA 2080Ti GPU and summarize the result in the table below. Note that these experiments are only used to test sampling efficiency, and the network weights are just randomly initialized. For fair comparison and to rule out the other factors other than different modeling space, we compare with ProtDiff and our LatentDiff without downsampling (named LatentDiff-P), as denoising networks for these models are similar, and we also make the number of parameters to be similar for these models. From the result, we can see that the generation time of 1000 protein structures in the protein space is about 2.9 hours, while it only takes about 11 minutes to generate in the latent space and then map to the protein space. So reducing modeling space demonstrates potential usefulness in practice. The sampling time of LatentDiff scales linearly with the number of diffusion steps because diffusion steps are performed sequentially. Moreover, since we use a fully connected graph for the diffusion model, the number of edges will increase quadratically with the number of nodes. Thus, the sampling time scales quadratically with protein length (latent nodes), as increasing latent nodes will quadratically increase memory consumption.
>
>
> | Method | Parameters | Protein Length | Latent node | Diffusion steps | Time (hrs) | Speed (sec/sample) |
> | ------ | ----- | ----- | ----- | ----- | ----- | ----- |
> | `ProtDiff` | 1974528 | 128 | N/A | 1000 | 1.9 | 6.85 |
> | `LatentDiff-P` | 2016453 | 128 | N/A | 1000 | 2.9 | 10.66 |
> | `LatentDiff` | 2027984 | 128 | 32 | 1000 | 0.18 | 0.68 |
> | `LatentDiff` | 2027984 | 128 | 32 | 2000 | 0.36 | 1.33 |
> | `LatentDiff` | 2027984 | 256 | 64 | 1000 | 0.73 | 2.66 |
>
>
> >**Q8: Broader Impact Concerns: The capacity to generate biological agents and filter them with computational approaches to select potentially dangerous agents should be considered.**
>
> - Thanks for the suggestion. We added "Broader Impact Statement" in the revision Section 6. We also copy it here:
>
>   - Our protein generation method, enabling the production of novel proteins, has a significantly broader impact potential. On one hand, it might offer potential opportunities for advancements in medicine, agriculture, and biotechnology, facilitating the development of innovative therapeutics, enzymes, and biomaterials in the future. On the other hand, while considering the concerns raised regarding the computational selection of potentially dangerous agents, we should prioritize responsible research practices, with stringent safety protocols, adherence to regulations, and collaboration with biosecurity experts to ensure the responsible handling of generated proteins. By fostering collaboration and knowledge dissemination, we aim to advance protein design while actively managing any potential risks associated with our method.

---

### Author Response · Authors · 2023-06-20
**General response**

We would like to thank the reviewers for their time and valuable comments on our work. We revised the manuscript heavily with additional experiments about efficiency according to the suggestions of reviewers and provided point-to-point responses to questions of reviewers in separate replies. Please let us know if there is any further concern or questions.

We summarize the main modifications of the manuscript below.

- Added one paragraph to discuss RFdiffusion and Chroma in Section 2.3.
- Added citation of two equivariant papers mentioned by reviewer NsUN in Section 2.3.
- Added motivation for reducing modeling space in Section 3.1.
- Modified overview part of Section 3.2 to clarify some designs of equivariant protein autoencoder.
- Moved secondary structure analysis experiment from Appendix to Section 4.5.
- Moved diversity experiment from Appendix to Section 4.6.
- Rewrote the efficiency comparison section (now Section 4.7), including clarification on parallel sampling efficiency and more experiments about efficiency.
- In the limitation part of Section 5, we analyzed why jointly generating 3D structures and sequences does not work better than predicting sequences from generated structures and added some suggestions about how to improve this in the future direction part of Section 5. In addition, we listed generating longer proteins and conditional generation in the future direction part of Section 5.
- Added "Broader Impact Statement" in Section 6, as suggested by reviewer NsUN.
- Modified Section A.2 about the dataset to make it more clear and avoid confusion.

---

### Decision · Action_Editors · 2023-07-19

**Recommendation:** Reject

**Comment:**

Thank you for submitting your manuscript to TMLR. We have carefully considered the feedback provided by three reviewers and have reached a decision regarding your paper.

Reviewer 1 has expressed a preference for acceptance, acknowledging the paper's contribution to the discussion around the use of diffusion models for protein structures and the technical novelty of the approach. We noticed that the results are not as strong as recent work by high-profile labs in this area and understood the lack of access to computational resources could have influenced the outcomes.

Reviewer 2 expressed the concerns about the fairness of the comparison with previous baselines due to the larger training data potentially providing an undue advantage. They strongly recommend conducting an ablation study on dataset sizes for LatentDiff to ensure a fair assessment of its performance relative to existing methods. We noticed some discussions between Reviewer 2 and the authors, and also observed the strong willingness of the authors to enhance their experiments according to the feedback (however, we understand this will take some time).

Reviewer 3 stated that the paper's results are not strong enough compared to previous research in the field and noting that the key idea (latent diffusion) is not very new.  But we also noticed that it is unfair to require people in the academia to have access to huge computational resources to compete with industrial labs in terms of model size and model performance.  We also noticed that the application of latent diffusion to protein generation has good implications to the related fields.

Based on the reviewers' comments, we believe that we cannot accept your paper in its current form, and we encourage you to majorly revise your paper before making a new submission. The revisions should include:
1) Conduct an ablation study on dataset sizes for LatentDiff to ensure a fair comparison with existing methods.
2) Provide a more in-depth discussion of the novelty of the latent diffusion approach and its contribution to the field.

**Audience:**

Using diffusion models for protein generation is definitely a trending topic, and many people in the TMLR community should be willing to see papers on this topic.

**Claims And Evidence:**

The paper has certain technical novelty, and the use of latent diffusion for protein generation has good implications to the related research fields. The experimental results can partly support the claims made by the authors, however, as pointed out by the reviewers, the current experiments are not solid enough (e.g., the comparison with baseline methods is somehow unfair and no ablation study is presented). There is the risk that more experiments or ablation studies might not be supportive to the major claim of the paper.

**Resubmission Of Major Revision:**

The authors may consider submitting a major revision at a later time.